# ALLOCATION OF PARAMETERS IN TRANSFORMERS

## ABSTRACT

Transformers have achieved remarkable successes across a wide range of applications, yet the theoretical foundation of their model efficiency remains underexplored. In this work, we investigate how the model parameters –mainly attention heads and head dimensions– should be allocated across layers to balance expressivity and efficiency. We first provide mathematical analysis on the role of early layers in information extraction from an approximation perspective, with a theoretical characterization on the trade-off between the number of heads and head dimension under a fixed parameter budget. In addition, we uncover and prove the *saturation* behavior of softmax activations: Continuously increasing head dimensions can lead to diminishing returns in learning errors, particularly for long sequences. Supported by both theory and experiments, this saturation pattern suggests that later layers can operate more efficiently with reduced parameters. Combining these insights, we propose principled strategies for allocating attention heads and dimensions across Transformers' layers, shedding light on theoretically-grounded model efficiency of Transformer-based architectures.

## 1 INTRODUCTION

Transformers have achieved remarkable success across natural language processing (Devlin et al., 2019; Brown et al., 2020; Grattafiori et al., 2024), computer vision (Dosovitskiy et al., 2021; Liu et al., 2021; Wu et al., 2021), and beyond, serving as the backbone of modern large language models (LLMs) and multi-modal systems due to their ability to capture long-range dependencies and scale efficiently (Vaswani et al., 2017). Their impact spans healthcare, supporting diagnostics and research (Meng, 2024; Busch, 2025); robotics, enabling natural language instruction following (Zeng et al., 2023; Liu et al., 2025); and chemistry, aiding molecular design and synthesis optimization (Bran et al., 2024; Jablonka et al., 2024).

Despite the popularity in practice, the theoretical understanding of why Transformers are effective, and more crucially, whether current parameter allocation (e.g., how to assign the number of heads and head dimensions per head, given fixed total budgets of model dimensions) is efficient, still remains limited. That is, while probing studies and interpretability analyzes have begun to shed light on the functions of different layers and heads (Rogers et al., 2020; Meng et al., 2022) in Transformers, a rigorous framework to explain their representational efficiency and parameter trade-offs is still lacking. This motivates principled investigations into how the parameters of Transformers (mainly attention heads and head dimensions) should be allocated, if one aims to bridge practical performance with deeper theoretical foundations.

To this end, we start with parameter allocation on individual layers of Transformers. A growing body of work shows that Transformer layers exhibit a division of labor pattern: Lower layers primarily encode surface-level and token-specific features, middle layers capture increasingly abstract syntactic relations, and higher layers emphasize semantic and task-specific information (Tenney et al., 2019; Jawahar et al., 2019; Clark et al., 2019; Van Aken et al., 2019; Rogers et al., 2020). Complementing these findings, Chen et al. (2024) demonstrated a case study on sparse linear regression, showing that multi-head attention is heavily activated in the first layer for input preprocessing, whereas subsequent layers typically rely on only a single dominant head to perform simplified optimization, revealing a "preprocess-then-optimize" mechanism. To sum up, these studies suggest that Transformers distribute representational and computational resources unevenly across layers, with early layers dedicated to token-level information extraction, and later layers consolidating and refining this information for higher-level reasoning.

Based on these insights, we first analyze parameter allocation in low layers for information extraction by establishing the approximation error estimate. In our analysis and established bound, one component of the error decreases with larger head dimensions ("dim"), while another decreases with more heads ("head"). Given a fixed budget of total parameters (say, dim × head fixed), this naturally implies a trade-off between the number of heads and head dimensions. In addition, we further prove the saturation pattern in softmax activations: Continuously increasing head dimensions can lead to diminishing returns in learning errors, particularly for long sequences. This saturation pattern suggests that one can operate more efficiently with reduced parameters (head dimensions) without significantly degrading performance, particularly for later layers that empirically activate only a few head (Chen et al., 2024). Both theoretical results (approximation error estimate and softmax saturation) are numerically verified through multiple simulations under varied settings, leading to principled parameter allocation strategies.

Our main contributions are summarized as follows:

- We establish the approximation error estimate on the information extraction in early Transformer layers, which implies the parameter trade-off between the number of heads and head dimensions.

- We uncover and prove the saturation pattern in softmax activations, which enables further parameter reduction, particularly in later Transformer layers.

- Both theoretical results are numerically verified, leading to principled strategies for efficient parameter allocation of heads and dimensions across layers.

**Notations.** For $k \in \mathbb{N}_+$, let $[k] = \{1, 2, ..., k\}$. For a vector $v$ and $1 \le p \le \infty$, we denote the $\ell_p$-norm of $v$ by $\|v\|_{\ell_p}$. For a matrix $A = (a_{ij})$, we denote the spectral norm of $A$ by $\|A\|_2$, and $\|A\|_\infty$ denotes the maximum norm, i.e., the maximum absolute entry of $A$; $\|A\|_{2,\infty}$ is defined by first computing the $\ell_2$ norm of each column of $A$, and then taking the maximum among them. For an event $S$, we define the indicator $\mathbb{I}_{\{S\}}$, which equals to 1 when $S$ occurs and 0 otherwise. We use the standard big-O notation $\mathcal{O}$ to hide absolute positive constants. For a matrix $X$, we denote the $a$-th to $b$-th columns by $X_{a:b}$.

## 2 RELATED WORKS

**Approximation ability of Transformers.** Yun et al. (2019) is the first work that established the universal approximation property of Transformers. Subsequent studies analyzed the efficiency of Transformers in representing specific functions or tasks, including sparse functions (Edelman et al., 2022), targets with nonlinear temporal kernels (Jiang & Li, 2023), practical computer programs (Giannou et al., 2023), long but sparse memories (Wang & E, 2024), induction heads (Sanford et al., 2024a;b; Rajaraman et al., 2024), and memorization and reasoning (Chen & Zou, 2024). Moreover, several studies suggest that Transformers achieve in-context learning by effectively approximating gradient-based iterations across layers (Garg et al., 2022; Akyürek et al., 2022; Von Oswald et al., 2023; Mahankali et al., 2023; Bai et al., 2023; Shen et al., 2023). Other work has examined limitations of Transformers' expressivity, particularly in modeling formal languages or simulating circuits (Hahn, 2020; Weiss et al., 2021; Bhattamishra et al., 2020; Merrill et al., 2022; Merrill & Sabharwal, 2023).

**Attention structure.** Several studies have analyzed the representational capabilities and structural properties of self-attention in Transformers. Likhosherstov et al. (2023) demonstrated that a fixed self-attention layer can approximate arbitrary sparse patterns with hidden size growing only logarithmically with the sequence length. Wang et al. (2020) and Choromanski et al. (2020) proposed substituting standard attention with linear structures for efficiency. Bhojanapalli et al. (2020) addressed the low-rank bottleneck in attention heads by setting head size proportional to the input sequence length, independent of the number of heads. Amsel et al. (2024) showed that a single full-rank attention head can implement nearest-neighbor search for any sequence length, whereas exponentially many low-rank heads are needed even for short sequences. Kajitsuka & Sato (2023) showed that one-layer, single-head Transformers possess memorization capacity for finite samples, whereas Hahn (2020) proved that self-attention cannot model certain formal languages unless the number of layers or heads grows with input length.

**Property of softmax.** Recent work has increasingly questioned the ubiquity of softmax attention in Transformers and investigated its theoretical and practical limitations. Mudarisov et al. (Mudarisov et al., 2025) develop a rigorous framework with distance, geometric, and gradient bounds, showing that as sequence length grows, attention weights collapse toward uniformity, token separability saturates, and gradients become unstable—phenomena empirically observed in GPT-2. Saratchandran et al. (Saratchandran et al., 2024) argue that the strength of softmax arises primarily from implicit Frobenius norm regularization rather than its probabilistic interpretation, and demonstrate that polynomial activations can match its effectiveness. Zheng et al. (Zheng et al., 2025) propose SA-Softmax (e.g., $x \cdot \mathrm{softmax}(x)$) to counteract gradient vanishing, validating its benefits empirically. Yan et al. (Yan et al., 2025) further provide theoretical evidence that sigmoid attention may achieve higher sample efficiency than softmax. Complementing these theoretical perspectives, Nakanishi (Nakanishi, 2025) introduces Scalable-Softmax (SSMax), which preserves sharper attention distributions as context length increases and outperforms softmax in focusing on salient tokens. Together, these studies highlight the limitations of softmax and motivate deeper theoretical analyses as well as the development of alternative normalization schemes.

## 3 PRELIMINARIES

In this work, we focus on each single layer of the Transformer architecture to provide sufficient quantitative characterizations for parameter allocation, mainly how to decide the number of heads and head dimensions. The Transformer maps input sequences to output sequences as follows.

**Input sequence.** We assume that the input sequence has length $L$, and each token $\boldsymbol{x}_t$ ($t \in [L]$) is in $\mathbb{R}^d$, and the input matrix is $\boldsymbol{X} = (\boldsymbol{x}_t)_{t \in [L]} \in \mathbb{R}^{d \times L}$. Also, we assume the $\ell_2$ norm of tokens are uniformly bounded by $B$.

**Embedding layer.** For each token $\boldsymbol{x}_t$, we transform it into a $D$-dimensional vector through a linear mapping $\boldsymbol{x}_t^{(0)} = \boldsymbol{W}_E \boldsymbol{x}_t + \boldsymbol{b}_E$ ($\boldsymbol{W}_E \in \mathbb{R}^{D \times d}$, $\boldsymbol{b}_E \in \mathbb{R}^D$). We denote the embedded input matrix by $\boldsymbol{X}^{(0)} \in \mathbb{R}^{D \times L}$.

**Transformer layer.** The embeddings ($\boldsymbol{X}^{(0)}$) are then passed through $N$ Transformer layers, each composed of a multi-head self-attention (MHSA) layer and a feed forward network (FFN):

$$\boldsymbol{X}^{(n-\frac{1}{2})} = \boldsymbol{X}^{(n-1)} + \mathrm{MHSA}^{(n)}(\boldsymbol{X}^{(n-1)}), \quad n \in [N],$$
$$\boldsymbol{X}^{(n)} = \boldsymbol{X}^{(n-\frac{1}{2})} + \mathrm{FFN}^{(n)}(\boldsymbol{X}^{(n-\frac{1}{2})}), \quad n \in [N].$$

Here

$$\mathrm{MHSA}^{(n)}(\boldsymbol{X}^{(n-1)}) = \boldsymbol{W}_O^{(n)} \cdot \mathrm{Concat}\left( \left( \mathrm{Attn}^{(n,h)}(\boldsymbol{X}^{(n-1)}) \right)_{h=1}^{H} \right),$$

$$\mathrm{Attn}^{(n,h)}(\boldsymbol{X}^{(n-1)}) = \boldsymbol{W}_V^{(n,h)} \boldsymbol{X}^{(n-1)} \mathrm{softmax}\left( \left\langle \boldsymbol{W}_Q^{(n,h)} \boldsymbol{X}^{(n-1)}, \boldsymbol{W}_K^{(n,h)} \boldsymbol{X}^{(n-1)} \right\rangle + \boldsymbol{R}^{(n,h)} \right),$$

where $\boldsymbol{W}_K^{(n,h)}, \boldsymbol{W}_Q^{(n,h)}, \boldsymbol{W}_V^{(n,h)} \in \mathbb{R}^{d_h^{(n)} \times D}$, and $\boldsymbol{W}_O^{(n)} \in \mathbb{R}^{D \times D}$ are learnable parameters (weight matrices) corresponding to the key, query, value and output matrix of the $h$-th head at the $n$-th layer, and $\langle \boldsymbol{A}, \boldsymbol{B} \rangle = \boldsymbol{A}^\top \boldsymbol{B}$. The activation function $\mathrm{softmax}(\cdot)$ is performed in a column-wise sense, and $\mathrm{Concat}(\cdot)$ acts vertically concatenates $H$ matrices and the sum of their dimension is $H$. $\boldsymbol{R}^{(n,h)}$ denotes the relative positional encoding (RPE) matrix (Vaswani et al., 2017; Devlin et al., 2019; Shaw et al., 2018; Dai et al., 2019; Raffel et al., 2019; Su et al., 2021; Press et al., 2022; Huang et al., 2018; Yang et al., 2019), which satisfies $\boldsymbol{R}_{ij}^{(n,h)} = -\infty$ if $i < j$. For instance, in the Alibi RPE (Press et al., 2022), we have $\boldsymbol{R}_{ij}^{(n,h)} = \phi(i - j; p^{(n,h)})$, where $p^{(n,h)}$ collects learnable parameters and

$$\phi(z) = \begin{cases} -p \cdot z, & z \geq 0, \\ -\infty, & \text{otherwise.} \end{cases}$$

**Feed forward network.** The feed forward network (FFN) is applied token-wisely, which maps each column of $\boldsymbol{X}^{(n-\frac{1}{2})}$ from $\mathbb{R}^D$ to $\mathbb{R}^D$ in the first $N-1$ layers, and from $\mathbb{R}^D$ back to $\mathbb{R}^d$ in the $N$-th layer. According to Cybenko (1989); Hornik et al. (1989); E et al. (2019); Lin & Jegelka (2018); Li et al. (2021; 2019), the FFN has universal approximation property as the number of parameter increases.

## 4 PARAMETER TRADE-OFFS IN INFORMATION EXTRACTION

As is pointed out by former literature, lower Transformer layers primarily capture surface-level, token-specific features: Chen et al. (2024) show that lower Transformer layers serve to preprocess contextual data, effectively learning simple and token-specific patterns. Similarly, Chen et al. (2025) observe that in smaller models, lower layers provide basic feature representations for individual tokens, which are subsequently leveraged by higher layers.

In this section, we investigate parameter allocation in the lower Transformer layers, aiming to determine the optimal number of heads and head dimension — given a fixed parameter budget (i.e. fixed model/embedding dimension) — that minimizes the error for an information extraction task, which we formulate as learning linear combinations of tokens.

**Linear sequence modeling.** Assume that $\boldsymbol{x}_t$'s are bounded vectors with zero means and identity covariances, and their $\ell_2$ norm are upper bounded by $B$. We consider a sequence-to-sequence modeling framework in which both the input $\boldsymbol{X} = (\boldsymbol{x}_t)_{t\in[L]} \in \mathbb{R}^{d\times L}$ and output $\boldsymbol{Y} = (\boldsymbol{y}_t)_{t\in[L]} \in \mathbb{R}^{d\times L}$ are sequences of $L$-length. For information extraction, we set $\boldsymbol{y}_t = \mathrm{H}_t(\boldsymbol{X}) = \sum_{i=0}^{L} \boldsymbol{\rho}_i \boldsymbol{x}_{t-i}$ (with a zero padding on $\boldsymbol{x}_t$), where the target kernel $\boldsymbol{\rho}_i \in \mathbb{R}^{d\times d}$, and $\|\boldsymbol{\rho}_i\|_2$ decreases with $i$ (alternatively, the kernel may be rearranged).

Here, we formulate the surface-level information extraction task as learning to construct linear combinations of input tokens, typically represented as a convolutional form. For a single Transformer layer, Jiang et al. (2025) identify a trade-off phenomenon in this type of tasks: as illustrated in their experiments (see Figure 8 in Appendix F), the trade-off between the number of attention heads and learning errors emerges in sequence modeling tasks, with its manifestation depending on the type and strength of underlying memory structures captured by the convolution kernel $\boldsymbol{\rho}$. Their results demonstrate that parameter trade-offs consistently arise across a variety of memory structures (e.g., exponentially/polynomially decaying kernels, delta/Airy function kernels). However, in certain cases — such as exponentially and polynomially decaying kernels — these trade-offs may vanish when the memory strength varies. This section's results also provide a theoretical explanation for this phenomenon.

In this case, our hypothesis space consists of single-layer Transformers with a fixed parameter budget (i.e., the embedding/model dimension) $D$, which we denote by $\mathcal{T}^D$ (i.e. all single-layer Transformers with a fixed embedding dimension $D$). The approximation error is defined as

$$\mathcal{E}_D(\boldsymbol{X}) = \inf_{\mathrm{T}\in\mathcal{T}^D} \|\mathrm{H}_t(\boldsymbol{X}) - \mathrm{T}(\boldsymbol{X})\|_{2,\infty}.$$

In this formulation, the embedding (model) dimension $D$ serves as a fixed parameter budget. Thus, parameters must be allocated judiciously to learn linear combinations of the most informative tokens. Suppose information can be extracted from only the $M$ tokens with the largest norm $\|\boldsymbol{\rho}_i\|_2$. We assign multiple attention heads, partitioned into $M$ groups, with each group dedicated to one token. Under this setup, we establish the following high-probability approximation error estimate and derive the corresponding parameter allocation strategy:

**Theorem 4.1.** *To approximate the linear target* $\mathrm{H}_t(\boldsymbol{X}) = \sum_{i=0}^{\infty} \boldsymbol{\rho}_i \boldsymbol{x}_{t-i}$, *we employ* $M$ *groups of heads, where group* $m$ *contains* $H_m$ *heads each of the dimension* $d_m$. *Given the fixed model dimension* $D = \sum_{m=1}^{M} H_m \cdot d_m$, *with probability at least* $1 - \delta$, *we have*

$$\mathcal{E}_D(\boldsymbol{X}) \leq \sum_{m=1}^{M} \|\boldsymbol{\rho}_m\|_2 B \left((1+\varepsilon_\delta)\mathbb{I}_{\{d_m \leq d\}}\sqrt{1 - \frac{d_m}{d}} + \frac{1.3\,e^{0.02m}}{H_m}\right) + \sum_{k=M+1}^{L} \|\boldsymbol{\rho}_k\|_2 B, \quad (1)$$

*where*

$$\varepsilon_\delta = \sqrt{\frac{2\log{(2ML/\delta)}}{\min_m d_m}},$$

*and the first term in equation (1) equals zero when $d_m > d$.*

**Corollary 4.1** (Parameter allocation via trade-offs). *Under the same conditions of Theorem 4.1, the allocation of parameters can be achieved by solving the following optimization problem*

$$\min_{M, H_m, d_m} \quad \sum_{m=1}^{M} \|\boldsymbol{\rho}_m\|_2 B \left( \mathbb{I}_{\{d_m \leq d\}} \sqrt{1 - \frac{d_m}{d}} + \frac{1.3\, e^{0.02m}}{H_m} \right) + \sum_{k=M+1}^{L} \|\boldsymbol{\rho}_k\|_2 B, \tag{2}$$

$$\text{s.t.} \quad \sum_{m=1}^{M} H_m \cdot d_m = D.$$

Corollary 4.1 is a direct conclusion of Theorem 4.1. We provide the detailed proof of Theorem 4.1 in Appendix C, and briefly summarize the main ideas here.

1. Since the embedding dimension is limited, we first compress the information contained in each token, which yields the first term of the approximation error (i.e. equation (1)).

2. We then employ $M$ groups of heads, where group $m$ is responsible for extracting the $m$-th token ahead. The approximation error decreases as more heads are employed, corresponding to the second term of the approximation error.

3. Finally, since only the $M$ most informative tokens are extracted, which truncates the target at $M$, we derive the third term of the approximation error.

According to Theorem 4.1, on the one hand, setting a large head dimension is beneficial for preserving token information, reducing the first error term in equation (1); on the other hand, increasing the number of heads is favorable for approximating the extraction function, reducing the second error term in equation (1). However, since the total budget/number of parameters is fixed, one can not increase the number of heads and head dimensions simultaneously beyond certain constraints, which inevitably leads to a trade-off.

Clearly, solving the optimization problem (2) in Corollary 4.1 yields a principled strategy for parameter allocation. Next, we provide some specific tasks as illustrations.

**N-gram.** The $n$-gram task in language modeling involves predicting the next token in a sequence using the previous $n$ tokens as contexts (Katz, 1987; Kneser & Ney, 1995; Chelba et al., 2017; Buck et al., 2014; Wang & E, 2024). This task captures local statistical dependencies in texts and serves as a basic benchmark for language modeling. We take a simple 4-gram task as an example, in which

$$\boldsymbol{y}_t = \boldsymbol{x}_{t-1} + \boldsymbol{x}_{t-2} + \boldsymbol{x}_{t-3} + \boldsymbol{x}_{t-4}. \tag{3}$$

In this case, $\boldsymbol{\rho}_i = \mathbb{I}_{\{1 \leq i \leq 4\}} \boldsymbol{I}_{d \times d}$. Let $d = 8, D = 256$ and $B = 1$, then (2) becomes

$$\min_{M, H_m, d_m} \quad \sum_{m=1}^{M} \mathbb{I}_{\{d_m \leq 8\}} \sqrt{1 - \frac{d_m}{8}} + \sum_{m=1}^{M} \frac{1.3\, e^{0.02m}}{H_m} + (4 - M),$$

$$\text{s.t.} \quad \sum_{m=1}^{M} H_m \cdot d_m = 256, \quad M \leq 4.$$

We numerically found by search that the optimal solution is the optimal solution is attained at $M = 4$, $d_m = H_m = 8$ $(m = 1, 2, 3, 4)$. See further experimental validations in Figure 2.

**Induction head.** The induction head (IH) is a mechanism in Transformers that learns to copy patterns by attending from a repeated token to its earlier occurrence, enabling sequence continuation beyond training contexts (Olsson et al., 2021). Implementing an IH requires two Transformer layers (Sanford et al., 2024a;b; Rajaraman et al., 2024; Wang et al., 2025). In this construction, the first layer plays a role similar to an $n$-gram task, extracting linear combinations of tokens. This allows us to propose an allocation strategy, with details provided in Appendix E.1.

**Nonlinear sequence modeling.** Theorem 4.1 can be extended to nonlinear cases. Consider a continuous, time-homogeneous system with the following decomposition (Wang & Li, 2023; de Figueiredo, 1982):

**Lemma 4.1** (Volterra Series Decomposition). H *is a continuous time-homogeneous system with input $\boldsymbol{X}$ and output $\boldsymbol{Y}$, then H can be expanded in the Volterra series as follows*

$$\boldsymbol{y}_t = h_0 + \sum_{n=1}^{\infty} \sum_{\tau_1=0}^{\infty} \cdots \sum_{\tau_n=0}^{\infty} \mathrm{H}^{(n)}(\tau_1, \ldots, \tau_n)\big(\boldsymbol{x}_{t-\tau_1} \otimes \cdots \otimes \boldsymbol{x}_{t-\tau_n}\big), \tag{4}$$

*where $\otimes$ denotes the Kronecker product. In particular, we call the expansion order $n$ to be the series' order.*

Consider the approximation and trade-off results for the $n$-th term of its Volterra decomposition $H^{(n)}$:

**Theorem 4.2** (Trade-offs via parameter allocation; informal). *To approximate the target $\mathrm{H}_t(\boldsymbol{X}) = \sum_{\tau_1=0}^{\infty} \cdots \sum_{\tau_n=0}^{\infty} \mathrm{H}^{(n)}(\tau_1, \ldots, \tau_n)\big(\boldsymbol{x}_{t-\tau_1} \otimes \cdots \otimes \boldsymbol{x}_{t-\tau_n}\big)$, we employ $M$ groups of heads, where group $m$ contains $H_m$ heads each of dimension $d_m \leq d$. Given the model dimension $D = \sum_{m=1}^{M} H_m \cdot d_m$ is fixed, with probability at least $1 - \delta$, we have*

$$\mathcal{E}_D(\boldsymbol{X}) \leq \sum_{\tau_1=0}^{M} \cdots \sum_{\tau_n=0}^{M} \big\|\mathrm{H}^{(n)}(\tau_1, \ldots, \tau_n)\big\|_2 (1 + \varepsilon_\delta)^n \mathbb{I}_{\{\min_m d_m \leq d\}} \sqrt{1 - \big(\tfrac{\min_m d_m}{d}\big)^n}$$

$$+ \sum_{\tau_1=0}^{M} \cdots \sum_{\tau_n=0}^{M} \big\|\mathrm{H}^{(n)}(\tau_1, \ldots, \tau_n)\big\|_2 \Big[(B + \varepsilon_{\mathrm{Attn}})^n - B^n\Big] + \varepsilon_{\mathrm{H}} + \varepsilon_{\mathrm{FFN}}.$$

*where*

$$\varepsilon_\delta = \sqrt{\tfrac{2\log(2ML/\delta)}{\min_m d_m}}, \ \varepsilon_{\mathrm{Attn}} = \max_m \frac{1.3\, e^{0.02T}}{H_m},$$

*and $\varepsilon_{\mathrm{H}}$ is caused by truncation of $\mathrm{H}^{(n)}$, $\varepsilon_{\mathrm{FFN}}$ is caused by using FFN to implement Kronecker product.*

The formal result and detailed proofs are deferred in Appendix E.2.

## 5 SOFTMAX SATURATION AND PARAMETER REDUCTION

In this section, we turn to the middle and later layers, which capture abstract syntactic relations as well as semantic and task-specific information. Since these layers employ only a few heads, we aim to reduce the total parameter budget by decreasing the head dimension. This is provably guaranteed by the saturation pattern of softmax activations, meaning that continuously increasing head dimensions can lead to diminishing returns in learning errors, particularly for long sequences. This saturation pattern suggests that one can operate more efficiently with reduced parameters (head dimensions) without significantly degrading performance, particularly for later layers that empirically rely on only a few heads (Chen et al., 2024).

Let

$$\mathrm{Logits}_{d_H}(\boldsymbol{W}_Q, \boldsymbol{W}_K, \boldsymbol{X}) = \Big\langle \boldsymbol{W}_Q \boldsymbol{X}, \boldsymbol{W}_K \boldsymbol{X} \Big\rangle + \boldsymbol{R} \in \mathbb{R}^{L \times L},$$

$$\mathrm{AttnScore}_{d_H}(\boldsymbol{W}_Q, \boldsymbol{W}_K, \boldsymbol{X}) = \mathrm{softmax}\big(\mathrm{Logits}_{d_H}(\boldsymbol{W}_Q, \boldsymbol{W}_K, \boldsymbol{X})\big) \in \mathbb{R}^{L \times L},$$

with $\boldsymbol{X} \in \mathbb{R}^{D \times L}$, $\boldsymbol{W}_K, \boldsymbol{W}_Q \in \mathbb{R}^{d_H \times D}$, and $\boldsymbol{R} \in \mathbb{R}^{L \times L}$. For $l \in [L]$, let $\boldsymbol{e}_l$ be the vector with the $l$-th element as 1 and all other elements as 0. Then we have the following estimates on the column-wise softmax Jacobians.

**Theorem 5.1** (The saturation pattern of softmax). *Assume that $\|\mathrm{Logits}_{d_H}(\boldsymbol{W}_Q, \boldsymbol{W}_K, \boldsymbol{X})\|_\infty$ is bounded. Let $J_l = \frac{\partial(\mathrm{AttnScore}_{d_H} \boldsymbol{e}_l)}{\partial(\mathrm{Logits}_{d_H} \boldsymbol{e}_l)}$ be the column-wise Jacobian for $l \in [L]$. Then we have*

$$\|J_l\|_2 = \mathcal{O}\left(\frac{1}{L}\right), \quad l = 1, 2, \ldots, L,$$

*where $\mathcal{O}$ hides absolute constants depending on $\boldsymbol{W}_Q, \boldsymbol{W}_K$ and $B$.*

According to Theorem 5.1, the "slope" of softmax activations is quite flat, particularly for long sequences. This further imply the following corollary, which characterizes the approximation error when "compressing" attention score matrices with reduced head dimensions.

**Corollary 5.1.** *Under the same condition as Theorem 5.1, for any* $\mathrm{AttnScore}_{d_H}(\boldsymbol{W}_Q, \boldsymbol{W}_K, \boldsymbol{X})$ *and* $d_h \leq d_H$, *there exist* $\hat{\boldsymbol{W}}_K, \hat{\boldsymbol{W}}_Q \in \mathbb{R}^{d_h \times D}$ *such that for any* $l \in [L]$

$$\left\| (\mathrm{AttnScore}_{d_H}(\boldsymbol{W}_Q, \boldsymbol{W}_K, \boldsymbol{X}) - \mathrm{AttnScore}_{d_h}(\hat{\boldsymbol{W}}_Q, \hat{\boldsymbol{W}}_K, \boldsymbol{X}))\boldsymbol{e}_l \right\|_2 = \mathcal{O}\left( \frac{\Lambda_h^H}{L} \right), \quad (5)$$

*where* $\Lambda_h^H = \sum_{i=d_h+1}^{d_H} \sigma(\boldsymbol{W}_Q^{T\top} \boldsymbol{W}_K^T)$ *is the tail sum of singular values of original logits matrices, and $\mathcal{O}$ hides absolute constants depending on* $\boldsymbol{W}_Q, \boldsymbol{W}_K$ *and $B$.*

The detailed proofs of Theorem 5.1 and Corollary 5.1 are provided in Appendix D. We further empirically show that the logits norm corresponding to hided constants and $\Lambda_h^H$ grows slowly with increased sequence lengths $L$ in Figure 10 in Appendix F as follows. Specifically, Corollary 5.1 is derived by applying an SVD-based low-rank approximation to original logits matrices, together with Theorem 5.1. According to Corollary 5.1, for long sequences ($L \gg 1$), compressing original attention score matrices in large head dimensions with those in smaller head dimensions can lead to minor errors, potentially leading to a principled strategy for additional parameter reduction.

**Strategy for additional parameter reduction.** Corollary 5.1 shows that a student head with reduced dimensions can approximate a teacher head with larger dimensions by only incurring small errors, particularly for long sequences. Therefore, a reasonable practical strategy for additional parameter reduction can be motivated as follows: For a pretrained Transformer model, we fit appropriate number of attention heads with a lower-dimensional student heads, by (i) first initializing student parameters with truncated SVD of teacher parameters; (ii) further training student parameters. This procedure enables further model compression with enhanced model efficiency, as shown in Figure 5 together with Figure 6 in Section 6.

## 6 EXPERIMENTS

In this section, we present numerical verifications of former theoretical results, i.e. information extraction (Theorem 4.1) and parameter reduction (Theorem 5.1 and Corollary 5.1). Throughout this section, layer and head indices are numbered starting from 0, in accordance with the program.

### 6.1 PARAMETER TRADE-OFFS IN INFORMATION EXTRACTION

**Tradeoff trend (verification of Theorem 4.1).** We first numerically validate the parameter tradeoff in the approximation error estimate (i.e. the first equation of (2)). Assume that all $d_m$'s are equal and there is no truncation error, we take $\rho = 1$, $d = 16$ and $D = 128$. Then

$$\mathcal{E}_{128} = \sqrt{1 - \frac{8}{H}} + \frac{1.3 \, e^{0.02}}{H}$$

without concentration error. We plot $\mathcal{E}(128)$ in Figure 1, which clearly shows a parameter trade-off trend. The minimum of $\mathcal{E}_{128}$ is achieved near $H = 8$, which is consistent with former theoretical predictions.

**4-gram task example (verification of parameter allocation strategy).** We empirically validate the parameter allocation strategy on the 4-gram task (Equation 3). A single-layer Transformer encoder with sinusoidal positional encoding (Vaswani et al., 2017) was trained on a synthetic dataset (white noises), where each target at time step $t$ is the sum of the previous four inputs (with initial padding) plus Gaussian noises. With a fixed embedding dimension $D = 256$, we run experiments under multiple learning rates and random seeds, and report the results corresponding to the best-performed configuration as an estimate of the approximation error. As shown in Figure 2, the optimal allocation occurs at $M = 4$ with $d_m = H_m = 8$ ($m = 1, 2, 3, 4$), consistent with our theory and confirming the proposed strategy (Equation 2).

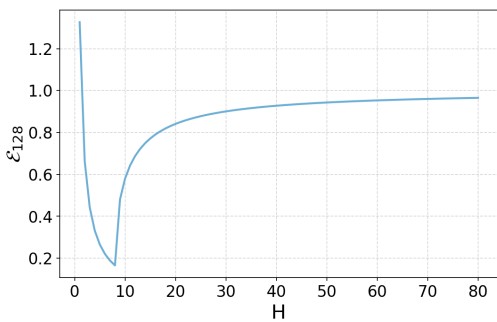

Figure 1: Trade-off Trend

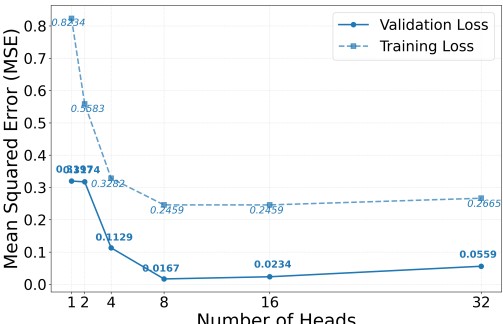

Figure 2: 4-gram Example

## 6.2 SOFTMAX SATURATION AND PARAMETER REDUCTION

**Spectral norm of Jacobian vs sequence length (verification of Theorem 5.1).** In Figure 3, we empirically verify Theorem 5.1 by analyzing how the spectral norm of the Jacobian matrix of the attention softmax output varies with sequence lengths in the `TinyLlama-1.1B-intermediate-step-1431k-3T` model. Specifically, we focus on Layer 0 and Head 0, using sequences from the Wikitext-103 dataset. For sequence lengths $L \in \{4, 8, 16, 32, 64, 128, 256, 512, 1024\}$, we sample 10 different sequences for each length.[1] For each sequence, we compute the Jacobian of the attention softmax output with respect to the logits, calculate its spectral norm, and average over the ten sequences, i.e., $\frac{1}{10}\sum_{l=1}^{10} \|J_l\|_2$ as in Theorem 5.1.

**Single head compression (verification of Corollary 5.1 and compression strategy).** We investigate low-head-dimension approximations of attention heads in pre-trained Transformer language models. Using the `ahxt/LiteLlama-460M-1T` model as the teacher, we isolated a single head (Layer 5, Head 0) and trained a student head with reduced query-key dimensions. The student retained the teacher's projection and rotary positional embeddings, with query and key projections parameterized as $D \times d_h$ matrices, where $d_h \in \{4, 8, 16, 24, 32, 48, 64\}$ and $D$ is the model embedding dimension. For $d_h < d_H$, the student projections were initialized via truncated SVD of the teacher's $W_Q^\top W_K$ matrix. The training objective was to fit the teacher head's attention weights, using mixed synthetic and WikiText-103 sequences of the length $1024$.[1] Optimization was performed with Adam (learning rate $10^{-3}$) for 10,000 epochs, and performance was evaluated by mean squared error on masked causal attention weights. The final learning error (equation 5) is plotted in Figure 4.

Our results demonstrate that low-dimensional student heads can effectively mimic the teacher's attention behavior with substantially fewer parameters. This provides empirical evidence for head compressibility and motivates strategies for parameter-efficient model design. In particular, a practical approach is to pretrain a Transformer model, then progressively replace each head with a smaller head dimension, enabling further model compression while improving parameter efficiency.

**Further verification in $6$-layer Transformer model.** In Figures 5 and 6, we conducted two complementary experiments to investigate Transformer training and attention head compressing. In the first experiment (Figure 5), we trained several 6-layer GPT-style Transformer decoders with varying head dimensions $d_H \in \{4, 8, 16, 24, 32, 48, 64\}$ on 1% of WikiText-103 ($\approx$1M tokens), employing RMSNorm, SwiGLU feed-forward networks, FlashAttention, and learned positional embeddings. The models, configured with 12 heads, were trained using AdamW (learning rate $3 \times 10^{-4}$, weight decay 0.1, cosine scheduler).

In the second experiment (Figure 6), we compress a single attention head (Layer 5, Head 0, $d_H = 64$) in the first experiment, which acts as a teacher, to student heads with reduced dimensions $d_h \in \{4, 8, 16, 24, 32, 48, 64\}$. The student heads retain the teacher's value projection and rotary

---

[1]Each sequence is generated by randomly selecting contiguous spans from Wikitext-103, which is tokenized to a fixed length $L$ and padded if necessary. For large $L$, padding ensures a consistent input size while maintaining meaningful contents.

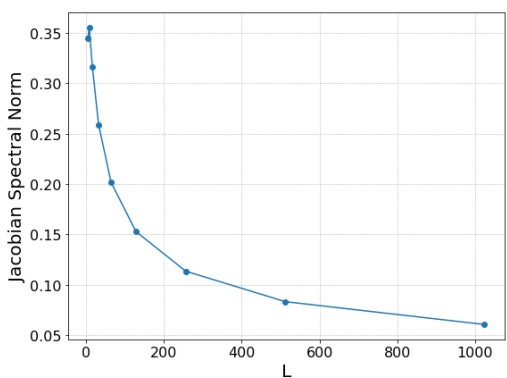

Figure 3: Saturation Scaling Law of Softmax

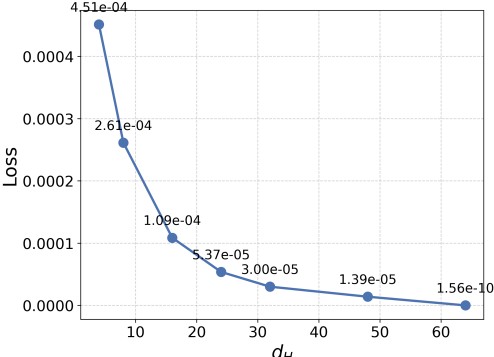

Figure 4: `LiteLlama` Single Head: Training Loss vs $d_h$

embeddings, while the query and key matrices were reparameterized with SVD-based initializations and optimized using mean squared error on attention weights.

Together, these experiments illustrate that Transformer parameter budgets can be reduced at multiple levels. At the full-model scale, a relatively small 6-layer decoder with reduced head dimensions achieves competitive performance, demonstrating that effective learning is possible with a limited parameter budget. At the level of individual attention heads, our compression results show that substantial dimensionality reduction (up to $4\times$ compression) incurs negligible performance degradation. These findings highlight both global and local opportunities for parameter-efficient Transformer design.

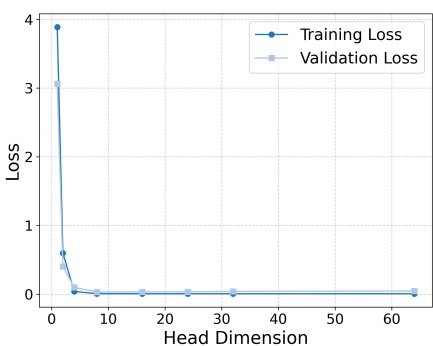

Figure 5: 6-layer Transformer: Loss vs Head Dimension

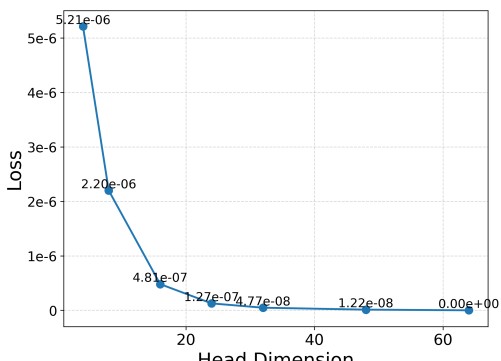

Figure 6: 6-layer Transformer Single Head: Training Loss v.s. Head Dimension

## 7 CONCLUSION

In this work, we studied how to allocate attention heads and head dimensions across Transformer layers under a fixed parameter budget. Our analysis reveals a trade-off between head number and dimension in early layers for token-level extraction, and shows a saturation effect in softmax activations, where larger head dimensions bring diminishing gains, especially on long sequences. These findings explain why later layers can remain efficient with fewer parameters and suggest strategies for designing compressed yet effective Transformer architectures. Future work includes extending the analysis to alternative attention mechanisms, exploring interactions with feed-forward modules and optimization, and validating the allocation strategies in large-scale pretraining and transfer, aiming to cut computation and memory costs without major performance loss.

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

# Appendix

## A  USE OF LARGE LANGUAGE MODELS

In the preparation of this manuscript, we employed a large language model (ChatGPT, developed by OpenAI) as a writing assistant. The model was used exclusively to:

- Improve the clarity, coherence, and conciseness of the text.
- Rephrase sentences to enhance grammatical correctness and readability.
- Ensure consistent terminology and smooth transitions across sections.

All outputs generated by the LLM were carefully reviewed, edited, and validated by the authors to guarantee technical accuracy, logical consistency, and fidelity to the research content. No text was included without human oversight. Importantly, the LLM was not employed for data generation, model training, or experimental design.

## B  LEMMAS USED IN PROOFS

The first lemma provides a convergence rate for approximating the delta function using exponential sums.

**Lemma B.1.** *For any $T \in \mathbb{N}_+$, $q, m \in \mathbb{N}_+$, there exists a $\phi_m^{\exp}(t) = \sum\limits_{k=1}^{m} \alpha_k e^{-\beta_k t}$ such that*

$$\|\mathbb{I}(\cdot = T) - \phi_m^{\exp}(\cdot)\|_{\ell_1(\mathbb{N})} \leq \frac{1.3\, e^{0.02T}}{m}$$

*where $\beta_k > 0$ holds for any $k \in [m]$, and $A, C > 0$ are absolute constants.*

*Proof.* This lemma is a corollary of Lemma F.1 in Wang & E (2024). By Lemma F.1 in Wang & E (2024) and its proof: for any $T \in \mathbb{N}_+$, $q, m \in \mathbb{N}_+$, there exists a $C(q) > 0$ and a $\phi_m^{\exp}(t) =$

$\sum\limits_{k=1}^{m} \alpha_k e^{-\beta_k t}$ such that

$$\|\mathbb{I}(\cdot = T) - \phi_m^{\exp}(\cdot)\|_{\ell_1(\mathbb{N})} \leq \frac{C(q)e^{0.01(q+1)T}}{m^q},$$

where $\beta_k > 0$ holds for any $k \in [m]$. Moreover,

$$C(q) = \frac{M(q)}{(1 - 1/e)^q}, \quad M(q) = \max_{0 \leq k \leq q} \sup_{x \in [-1,1]} \left| \Psi^{(k)}(x) \right|,$$

where $\Psi(x) = \begin{cases} \exp\left(-\frac{1}{1-x^2}\right), & x \in (-1, 1) \\ 0, & \text{otherwise} \end{cases}$ is the standard bump function on $[-1, 1]$. We mainly focus on small $q$ since $C(q)$ grows rapidly and through directly computation, we have $C(1) \approx 1.3$. Hence,

$$\|\mathbb{I}(\cdot = T) - \phi_m^{\exp}(\cdot)\|_{\ell_1(\mathbb{N})} \leq \frac{1.3 \, e^{0.02T}}{m}$$

$\square$

The following lemma guarantees that, with high probability, a bounded random vector can be projected onto a $d$-dimensional subspace such that the resulting error, after applying any linear transformation $A$, is tightly controlled by $\|A\|_2$, the projection dimension, and the vector's norm bound.

**Lemma B.2** (High-probability projection). *Let $\boldsymbol{x} \in \mathbb{R}^D$ be a random vector with zero mean, identity covariance $\mathbb{E}[\boldsymbol{x}\boldsymbol{x}^\top] = I_D$, and almost surely bounded norm $\|\boldsymbol{x}\|_2 \leq B$. Let $\boldsymbol{A} \in \mathbb{R}^{m \times D}$ be any fixed matrix. Then, for any $\delta \in (0, 1)$, there exists an orthogonal projection matrix $\boldsymbol{P} \in \mathbb{R}^{d \times D}$ such that with probability at least $1 - \delta$ over the randomness of $\boldsymbol{P}$, we have*

$$\|\boldsymbol{A}\boldsymbol{x} - \boldsymbol{A}\boldsymbol{P}^\top \boldsymbol{P}\boldsymbol{x}\|_2 \leq \|\boldsymbol{A}\|_2 \, B \, \sqrt{1 - \frac{d}{D}} \, \left(1 + \sqrt{\frac{2\log(2/\delta)}{d}}\right).$$

*Proof.* We construct $\boldsymbol{P} \in \mathbb{R}^{d \times D}$ as a random orthogonal projection (e.g., by taking the first $d$ rows of a Haar-distributed orthogonal matrix, or using a Johnson-Lindenstrauss style random matrix). Then

$$\|\boldsymbol{A}\boldsymbol{x} - \boldsymbol{A}\boldsymbol{P}^\top \boldsymbol{P}\boldsymbol{x}\|_2 = \|\boldsymbol{A}(I - \boldsymbol{P}^\top \boldsymbol{P})\boldsymbol{x}\|_2 \leq \|\boldsymbol{A}\|_2 \, \|(I - \boldsymbol{P}^\top \boldsymbol{P})\boldsymbol{x}\|_2.$$

From properties of random orthogonal projections, the squared projection error satisfies

$$\mathbb{E}\|(\boldsymbol{I} - \boldsymbol{P}^\top \boldsymbol{P})\boldsymbol{x}\|_2^2 = \|\boldsymbol{x}\|_2^2 \left(1 - \frac{d}{D}\right) \leq B^2 \left(1 - \frac{d}{D}\right).$$

Moreover, the projection error $\|(\boldsymbol{I} - \boldsymbol{P}^\top \boldsymbol{P})\boldsymbol{x}\|_2$ is a sub-Gaussian random variable with mean at most $B\sqrt{1 - d/D}$, so by a standard sub-Gaussian tail bound (Hoeffding inequality):

$$\Pr\left[\|(\boldsymbol{I} - \boldsymbol{P}^\top \boldsymbol{P})\boldsymbol{x}\|_2 \geq B\sqrt{1 - d/D}\,(1 + t)\right] \leq 2\exp\left(-\frac{dt^2}{2}\right).$$

Taking $t = \sqrt{\frac{2\log(2/\delta)}{d}}$ gives

$$\|(\boldsymbol{I} - \boldsymbol{P}^\top \boldsymbol{P})\boldsymbol{x}\|_2 \leq B\sqrt{1 - d/D} \left(1 + \sqrt{\frac{2\log(2/\delta)}{d}}\right)$$

with probability at least $1 - \delta$. Multiplying by $\|\boldsymbol{A}\|_2$ completes the proof. $\square$

The following lemma is Corollary 1 in Mudarisov et al. (2025).

**Lemma B.3.** *Assume $\|\text{Logits}_{d_H}(\boldsymbol{W}_Q, \boldsymbol{W}_K, \boldsymbol{X})\|_\infty$ is bounded, the attention weights satisfy:*

$$\frac{1}{L}\exp(-2\Delta) \le \|\text{AttnScore}_{d_H}(\boldsymbol{W}_Q, \boldsymbol{W}_K, \boldsymbol{X})\|_\infty \le \frac{1}{L}\exp(2\Delta),$$

*where $\Delta = \|\boldsymbol{W}_Q\|_2\|\boldsymbol{W}_K\|_2 B^2$.*

The Gershgorin circle theorem localizes all eigenvalues of a matrix within discs determined by its diagonal entries and row sums of off-diagonal magnitudes.

**Lemma B.4** (Gershgorin circle theorem). *Let $A = (a_{ij}) \in \mathbb{C}^{n \times n}$. For each $i$ define the Gershgorin radius*

$$R_i := \sum_{j \ne i} |a_{ij}|.$$

*Then every eigenvalue $\lambda$ of $A$ lies in the union of the Gershgorin discs*

$$\lambda \in \bigcup_{i=1}^{n} \{z \in \mathbb{C} : |z - a_{ii}| \le R_i\}.$$

*In particular, if the discs $\{z : |z - a_{ii}| \le R_i\}$ are contained in a region $S \subset \mathbb{C}$, then all eigenvalues of $A$ lie in $S$.*

The mean value theorem expresses the difference $F(\boldsymbol{y}) - F(\boldsymbol{x})$ as the Jacobian at an intermediate point applied to $\boldsymbol{y} - \boldsymbol{x}$.

**Lemma B.5** (Mean Value Inequality for vector-valued functions). *Let $F : \mathbb{R}^n \to \mathbb{R}^m$ be continuously differentiable on an open convex set $D \subset \mathbb{R}^n$. Then for any $\boldsymbol{x}, \boldsymbol{y} \in D$, there exists $\theta \in (0,1)$ such that*

$$\|F(\boldsymbol{y}) - F(\boldsymbol{x})\| \le \|JF(\boldsymbol{x} + \theta(\boldsymbol{y} - \boldsymbol{x}))\| \cdot \|\boldsymbol{y} - \boldsymbol{x}\|,$$

*where $JF(\cdot)$ denotes the Jacobian matrix of $F$ and $\|\cdot\|$ is an appropriate vector/matrix norm.*

## C  PROOF OF THEOREM 4.1

In this section, we give the detailed proof of Theorem 4.1. We first restate the theorem:

**Theorem C.1** (Restate of Theorem 4.). *To approximate the linear target $\text{H}_t(\boldsymbol{X}) = \sum_{i=0}^{\infty} \boldsymbol{\rho}_i \boldsymbol{x}_{t-i}$, we employ $M$ groups of heads, where group $m$ contains $H_m$ heads each of dimension. Given the model dimension $D = \sum_{m=1}^{M} H_m \cdot d_m$ is fixed, with probability at least $1 - \delta$, we have*

$$\mathcal{E}_D(\boldsymbol{X}) \le \sum_{m=1}^{M} \|\boldsymbol{\rho}_m\|_2 B \left( (1 + \varepsilon_\delta)\mathbb{I}_{\{d_m \le d\}}\sqrt{1 - \frac{d_m}{d}} + \frac{1.3\, e^{0.02m}}{H_m} \right) + \sum_{k=M+1}^{L} \|\boldsymbol{\rho}_k\|_2 B, \quad (6)$$

*where*

$$\varepsilon_\delta = \sqrt{\frac{2\log(2ML/\delta)}{\min_m d_m}},$$

*and the first term in equation (1) takes zero when $d_m > d$.*

*Proof.* We only consider the case $d_m \le d$ in our proof, since otherwise there is no information loss and the first term in the approximation error vanishes. Since we only use a single layer of the Transformer, we omit the layer index.

**Embedding layer.** We embed each token by

$$\boldsymbol{W}_E = \begin{pmatrix} \boldsymbol{I}_d \\ \boldsymbol{0}_{(D-d) \times d} \end{pmatrix} \in \mathbb{R}^{D \times d}, \quad \boldsymbol{b}_E = \boldsymbol{0} \in \mathbb{R}^D,$$

for $\boldsymbol{x}_t$, then each token $\boldsymbol{x}_t^{(0)}$ after embedding is

$$\boldsymbol{x}_t^{(0)} = \boldsymbol{W}_E \boldsymbol{x}_t + \boldsymbol{b}_E = \begin{pmatrix} \boldsymbol{x}_t \\ \boldsymbol{0} \end{pmatrix} \in \mathbb{R}^D.$$

**Multi-head attention layer with residual connection.** We implement multi-head attention using $M$ groups of heads. Each group $m$ comprises $H_m$ heads with dimension $d_m$ for $(m = 1, 2, \ldots, M)$, aimed at extracting the corresponding representation $\tilde{\boldsymbol{x}}_{t-m}$ which is $\boldsymbol{x}_{t-m}$ after projection to $d_m$-dimension space. This step can be presented as follows for each token:

$$
\begin{pmatrix} \boldsymbol{x}_t \\ 0 \\ \vdots \\ 0 \end{pmatrix} \longrightarrow \begin{pmatrix} \tilde{\boldsymbol{x}}_t \\ \tilde{\boldsymbol{x}}_{t-1} \\ \vdots \\ \tilde{\boldsymbol{x}}_{t-M} \\ \boldsymbol{0}_{D-(\sum_{m=1}^M d_m)} \end{pmatrix} := \boldsymbol{x}_t^{(1/2)}.
$$

where $\tilde{\boldsymbol{x}}_t$ is obtained by residual connection.

We now focus on the extraction of an arbitrary $\tilde{\boldsymbol{x}}_{t-m}$ and give the details of construction.

By lemma B.1, for any rate $q \in \mathbb{N}^+$, there exists a function

$$
\phi_m^{\exp}(t) = \sum_{h=1}^{H_m} \alpha_{h,m} e^{-\beta_{h,m}(t-1)}
$$

such that $\beta_h > 0$ and

$$
\|\mathbb{I}\,(\cdot = m) - \phi_m^{\exp}(\cdot)\|_{\ell_1(\mathbb{N})} = \sum_{s=0}^{+\infty} |\mathbb{I}\,(s = m) - \phi^{\exp}(s)| \leq \frac{1.3\,e^{0.02m}}{H_m}.
$$

By Lemma B.2, there exist an projection $\boldsymbol{P}_m \in \mathbb{R}^{d_m \times d}$ such that

$$
\Pr\left[ \|(\boldsymbol{I} - \boldsymbol{P}_m^\top \boldsymbol{P}_m)\boldsymbol{x}_l\|_2 \geq B\sqrt{1 - \frac{d_m}{d}}\,(1 + \varepsilon) \right] \leq 2\exp\left(-\frac{d_m \varepsilon^2}{2}\right).
$$

for any $l = 1, \ldots, L$. Let $\tilde{\boldsymbol{x}}_{t-m} := \boldsymbol{P}_m \boldsymbol{x}_{t-m}$ For $h = \sum_{i=1}^{m-1} H_i, \sum_{j=1}^{m-1} H_j + 1, \ldots, \sum_{k=1}^m H_k$, we choose parameters as follows

$$
p^{(h)} = \beta_{h,m}, \quad \boldsymbol{W}_V^{(h)} = \alpha_{h,m}\left(\sum_{i=0}^{H_m} \exp(-\beta_{h,m}(i-1))\right)\boldsymbol{P}_m, \quad \boldsymbol{W}_K^{(h)} = \boldsymbol{W}_Q^{(h)} = 0.
$$

The output of $H$ heads are concatenated together, and we take

$$
\boldsymbol{W}_O = \sum_{m=1}^M \sum_{h=\sum_{i=1}^{m-1} H_i+1}^{\sum_{j=1}^m H_j} \boldsymbol{S}_{m,h},
$$

where $\boldsymbol{S}_{m,h} \in \mathbb{R}^{D \times D}$ moves the output of the $m-$th set of heads to the same position

$$
\boldsymbol{S}_{m,h} = \begin{pmatrix} \boldsymbol{0}_{(\sum_{i=1}^{m-1} d_i)\times(\sum_{i=1}^{m-1} d_i+(h-1)d_m)} & \boldsymbol{0}_{(\sum_{i=1}^{m-1} d_i)\times d_m} & \boldsymbol{0}_{(\sum_{i=1}^{m-1} d_i)\times(D-\sum_{i=1}^{m-1} d_i-hd_m)} \\ \boldsymbol{0}_{d_m\times(\sum_{i=1}^{m-1} d_i+(h-1)d_m)} & \boldsymbol{I}_{d_m\times d_m} & \boldsymbol{0}_{d_m\times(D-\sum_{i=1}^{m-1} d_i-hd_m)} \\ \boldsymbol{0}_{(D-\sum_{i=1}^m d_m)\times(\sum_{i=1}^{m-1} d_i+(h-1)d_m)} & \boldsymbol{0}_{(D-\sum_{i=1}^m d_m)\times d_m} & \boldsymbol{0}_{(D-\sum_{i=1}^m d_m)\times(D-\sum_{i=1}^{m-1} d_i-hd_m)} \end{pmatrix}
$$

We define

$$
\tilde{\boldsymbol{P}}_m := \left(0_{d_H \times (m-1)d_H}, I_{d_m \times d_m}, 0_{d_H \times (D-md_H)}\right) \in \mathbb{R}^{d_m \times D}, \quad m \in [M],
$$

then

$$
\tilde{\boldsymbol{P}}_m \cdot \mathrm{Concat}\left(\mathrm{Attn}^{(h)}(\boldsymbol{X}^{(0)})\right)_{h=1}^H = \sum_{h=\sum_{i=1}^{m-1} H_i}^{\sum_{j=1}^m H_j} \alpha_{h,m} \sum_{s=1}^\infty e^{-\beta_{h,m}(s-1)} \tilde{\boldsymbol{x}}_{t-s} := \hat{\boldsymbol{x}}_{t-m}.
$$

And the approximation error of this step is

$$\varepsilon_{\text{Attn}} \le \sup_t \sum_{m=1}^M \|\tilde{\boldsymbol{x}}_{t-m} - \hat{\boldsymbol{x}}_{t-m}\|_\infty$$

$$\le \sup_t \sum_{m=1}^M \|\mathbb{I}(\cdot = m) - \phi_m^{\exp}(\cdot)\|_{\ell_1(\mathbb{N})} \cdot B$$

$$\le B \cdot \sum_{m=1}^M \frac{1.3\, e^{0.02m}}{H_m},$$

**Feed forward network.** FFN is implemented component-wise to realize convolution. The final output for the $t$-th token is

$$\hat{H}_t(\boldsymbol{X}) = \sum_{m=0}^M \tilde{\boldsymbol{\rho}}_m \hat{\boldsymbol{x}}_{t-m},$$

where $\tilde{\boldsymbol{\rho}}_m = \boldsymbol{\rho}_m \boldsymbol{P}_m^\top$. In fact we only need a linear map to achieve this, and FFN won't cause any approximation error.

**Approximation error.** By Lemma B.2, with probability at least $1 - M \max_m 2 \exp\left(-\frac{d_m \varepsilon^2}{2}\right)$, we have

$$\mathcal{E}_D(\boldsymbol{X}) = \|\sum_{m=0}^\infty \boldsymbol{\rho}_m \boldsymbol{x}_{t-m} - \sum_{m=0}^M \tilde{\boldsymbol{\rho}}_m \hat{\boldsymbol{x}}_{t-m}\|_2$$

$$\le \sum_{m=1}^M \|\boldsymbol{\rho}_m \boldsymbol{x}_{t-m} - \tilde{\boldsymbol{\rho}}_m \hat{\boldsymbol{x}}_{t-m}\|_2 + \sum_{k=M+1}^\infty \|\boldsymbol{\rho}_k\|_2 B$$

$$\le \sum_{m=1}^M \|\boldsymbol{\rho}_m \boldsymbol{x}_{t-m} - \tilde{\boldsymbol{\rho}}_m \tilde{\boldsymbol{x}}_{t-m}\|_2 + \sum_{m=1}^M \|\tilde{\boldsymbol{\rho}}_m \tilde{\boldsymbol{x}}_{t-m} - \tilde{\boldsymbol{\rho}}_m \hat{\boldsymbol{x}}_{t-m}\|_2 + \sum_{k=M+1}^\infty \|\boldsymbol{\rho}_k\|_2 B$$

$$(\text{By Lemma B.2}) \le B \sum_{m=1}^M \|\boldsymbol{\rho}_m\|_2 \left(\sqrt{1 - \frac{d_m}{d}}(1+\varepsilon) + \varepsilon_{\text{Attn}}\right) + B \sum_{k=M+1}^\infty \|\boldsymbol{\rho}_k\|_2.$$

Take uniform bound over $L$ tokens, with probability at least $1 - \delta$, we have

$$\mathcal{E}_D(\boldsymbol{X}) \le \sum_{m=1}^M \|\boldsymbol{\rho}_m\|_2 B \left(\sqrt{1 - \frac{d_m}{d}}\left(1 + \sqrt{\frac{2\log(2ML/\delta)}{\min_m d_m}}\right) + \varepsilon_{\text{Attn}}\right) + \sum_{k=M+1}^\infty \|\boldsymbol{\rho}_k\|_2 B$$

$$\square$$

# D    PROOF OF THEOREM 5.1 AND COROLLARY 5.1

In this section, we prove Theorem 5.1 and Corollary 5.1, we first restate the results:

**Theorem D.1** (Restate of Theorem 5.1). *Assume* $\|\text{Logits}_{d_H}(\boldsymbol{W}_Q, \boldsymbol{W}_K, \boldsymbol{X})\|_\infty$ *is bounded. Let* $J_l = \frac{\partial(\text{AttnScore}_{d_H} \boldsymbol{e}_l)}{\partial(\text{Logits}_{d_H} \boldsymbol{e}_l)}$ *be the column-wise Jacobian for* $l \in [L]$. *Then we have*

$$\|J_l\|_2 = \mathcal{O}\left(\frac{1}{L}\right), \quad l = 1, 2, \ldots, L,$$

*where* $\mathcal{O}$ *hides an absolute constant depends on* $\text{Logits}_{d_H}(\boldsymbol{W}_Q, \boldsymbol{W}_K, \boldsymbol{X})$.

**Corollary D.1** (Restatement of Corollary 5.1). *Under the same condition as Theorem 5.1, for any* $\text{AttnScore}_{d_H}(\boldsymbol{W}_Q, \boldsymbol{W}_K, \boldsymbol{X})$ *and* $d_h \leq d_H$, *there exist* $\hat{\boldsymbol{W}}_K, \hat{\boldsymbol{W}}_Q \in \mathbb{R}^{d_h \times D}$ *such that for any* $l \in [L]$

$$\left\| (\text{AttnScore}_{d_H}(\boldsymbol{W}_Q, \boldsymbol{W}_K, \boldsymbol{X}) - \text{AttnScore}_{d_h}(\hat{\boldsymbol{W}}_Q, \hat{\boldsymbol{W}}_K, \boldsymbol{X}))\boldsymbol{e}_l \right\|_2 = \mathcal{O}\left( \frac{\Lambda_h^H}{L} \right), \quad (7)$$

*where* $\Lambda_h^H = \sum_{i=d_h+1}^{d_H} \sigma(\boldsymbol{W}_Q^{T\top} \boldsymbol{W}_K^T)$ *is the tail sum of singular values of original logits matrices, and* $\mathcal{O}$ *hides an absolute constant depends on* $\text{Logits}_{d_H}(\boldsymbol{W}_Q, \boldsymbol{W}_K, \boldsymbol{X})$.

*Proof.* Without loss of generality, we focus on a certain index $l$. For simplicity of notation, we let $\boldsymbol{z}$ denote the $l$-th column of $\left\langle \boldsymbol{W}_Q \boldsymbol{X}, \boldsymbol{W}_K \boldsymbol{X} \right\rangle + \boldsymbol{R}$ and denote softmax as $\sigma$, and the $i$-th component of output vector as $\sigma_i(\boldsymbol{z})$.

Through direct computation, we have

$$J\sigma(\boldsymbol{z}) = diag(\sigma(\boldsymbol{z})) - \sigma(\boldsymbol{z})\sigma(\boldsymbol{z})^\top = \begin{cases} \sigma_i(\boldsymbol{z}) - \sigma_i(\boldsymbol{z})^2, & i = j, \\ -\sigma_i(\boldsymbol{z})\sigma_j(\boldsymbol{z}), & i \neq j. \end{cases}$$

And notice that $J\sigma(\boldsymbol{z})$ is semidefinite. In fact, let $p = \sigma(\boldsymbol{z}) \in \mathbb{R}^L$, so that $p_i \geq 0$ and $\sum_{i=1}^L p_i = 1$. For any $v \in \mathbb{R}^L$, we compute the quadratic form:

$$v^\top J\sigma(\boldsymbol{z})v = v^\top \left( \text{diag}(p) - pp^\top \right)v = \sum_{i=1}^L p_i v_i^2 - \left( \sum_{i=1}^L p_i v_i \right)^2.$$

The right-hand side is exactly the variance of the random variable $V$ taking values $v_i$ with probability $p_i$, i.e.

$$v^\top J\sigma(\boldsymbol{z})v = \mathbb{E}_p[V^2] - \left( \mathbb{E}_p[V] \right)^2 = \text{Var}_p(V) \geq 0.$$

Therefore $J\sigma(\boldsymbol{z})$ is positive semidefinite.

According to Lemma B.4, for an eigenvalue $\lambda$ of $J\sigma(\boldsymbol{z})$, there exists an $i$ such that

$$|\lambda - \sigma_i(\boldsymbol{z})(1 - \sigma_i(\boldsymbol{z}))| \leq \sum_{j \neq i} \sigma_i(\boldsymbol{z})\sigma_j(\boldsymbol{z}) = \sigma_i(\boldsymbol{z})(1 - \sigma_i(\boldsymbol{z})).$$

Hence, $0 \leq \lambda \leq \frac{1}{2}$.

Since the sum of the eigenvalues of a matrix is equal to its trace, we have

$$\sum_i \lambda_i \leq \sum_i \sigma_i(\boldsymbol{z})(1 - \sigma_i(\boldsymbol{z})) \leq 1.$$

By Lemma B.3, $\sigma_i(\boldsymbol{z}) \sim \frac{1}{L}$, and $\lambda_i \sim \frac{1}{L}$ for $i = 1, 2, \ldots, L$. Hence, $\|J\sigma(\boldsymbol{z})\| \sim \frac{1}{L}$.

To prove Corollary 5.1, we take $\hat{\boldsymbol{W}}_Q^\top \hat{\boldsymbol{W}}_K$ as the low-rank approximation of $\boldsymbol{W}_Q^\top \boldsymbol{W}_K$. then Corollary 5.1 can be dirctectly obtained from Theorem 5.1 and Lemma B.5.

$\square$

# E APPROXIMATION AND TRADEOFF RESULTS FOR NONLINEAR TARGET

## E.1 INDUCTION HEAD

Wang et al. (2025) defined a generalized induction head:

$$\text{IH}_{n:t}(\boldsymbol{X}) = \sum_{s=n}^{t-1} \pi_s \boldsymbol{x}_s,$$

where

$$\pi_s = \mathrm{softmax}\left( (\boldsymbol{X}_{t-n+2:t} \boldsymbol{W}^* \boldsymbol{X}_{v-n+1:v-1})_{v=n}^{L-1} \right)_{v=s},$$

and empirically pointed out that the first Transformer layer is used to extract information. In fact, the parameter allocation problem for the first layer reduces to the $n$-gram case.

We systematically investigated the effect of Transformer architectural variants on the Induction Head task in Figure 7, which tests a model's ability to recognize and extend repeated patterns—a fundamental reasoning mechanism for in-context learning. The task was constructed using synthetic sequences of length 128 with 10 unique tokens and embedded 5-token repeating patterns, yielding a dataset of 10,000 sequences (80%/20% train/validation).

The model architecture was a two-layer Transformer encoder with $d_{\mathrm{model}} = 256$, pre-norm configuration, GELU feed-forward networks, learned positional encodings, and a two-layer MLP output head. We varied the number of attention heads in $1, 2, 4, 8, 16, 32$ while keeping $d_{\mathrm{model}}$ fixed. Models were trained for 500 epochs using AdamW (learning rate $\in 10^{-3}, 5! \times !10^{-4}, 10^{-4}$, weight decay $10^{-4}$), cosine scheduling with ReduceLROnPlateau, gradient clipping (1.0), and mixed precision. Each configuration was repeated with three random seeds, and the best-performing model was selected by validation mean squared error (MSE).

Evaluation included validation MSE, heatmaps of performance across architectural variants, and qualitative prediction analysis. The results reveal how the number of heads influences the ability of the model to capture the Induction Head mechanism and demonstrate the robustness of certain configurations across seeds. This provides practical guidance for architectural design in tasks requiring in-context learning.

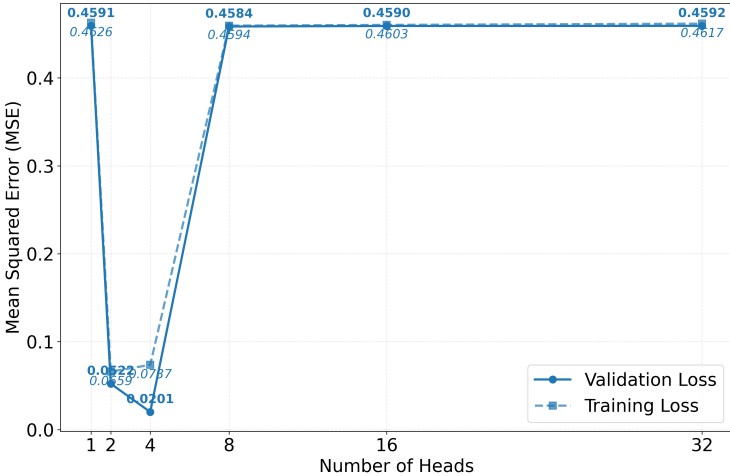

Figure 7: Induction Head Tradeoff

## E.2 Continuous Time-homogeneous System

In this subection, we analyze the approximation error of a continuous target function and derive the resulting trade-off.

We consider a sequence-to-sequence model in which both the input $\boldsymbol{X} = (\boldsymbol{x}_t)_{t\in\mathbb{Z}} \subset \mathcal{X} \subset \mathbb{R}^{d\times\mathbb{Z}}$ and output $\boldsymbol{Y} = (\boldsymbol{y}_t)_{t\in\mathbb{Z}} \subset \mathcal{Y} \subset \mathbb{R}^{d\times\mathbb{Z}}$ are sequences of infinite length and input tokens $\boldsymbol{x}_t$ are i.i.d. (we need a stronger assumption due to the target) random vectors with zero mean and identity covariance matrix and whose norm is bounded by $B$. $H$ denotes the mapping from $\mathcal{X}$ to $\mathcal{Y}$ with $\boldsymbol{y}_t = \mathrm{H}_t(\boldsymbol{X})$.

To give the mapping H practical relevance, we must impose certain properties on the target function space. We now present the definitions of these properties.

**Definition E.1.** $\mathrm{H} = \{\mathrm{H}_t : \mathcal{X} \mapsto \mathbb{R}^d; t \in \mathbb{R}\}$ be a sequence of functionals.

1.(**Linear**) $H_t$ is linear if for any $\lambda, \lambda' \in \mathbb{R}$ and $\boldsymbol{X}, \boldsymbol{X}' \in \mathcal{X}$, $H_t(\lambda \boldsymbol{X} + \lambda' \boldsymbol{X}') = \lambda H_t(\boldsymbol{X}) + \lambda H_t(\boldsymbol{X}')$.

2.(**Continuous**) $H_t$ is continuous if for any $\boldsymbol{X}, \boldsymbol{X}' \in \mathcal{X}$, $\lim_{\boldsymbol{X}' \to \boldsymbol{X}} |H_t(\boldsymbol{X}') - H_t(\boldsymbol{X})| = 0$.

3.(**Bounded**) $H_t$ is bounded if $\sup_{\boldsymbol{X} \in \mathcal{X}, \boldsymbol{X} \neq \boldsymbol{0}} \frac{|H_t(\boldsymbol{X})|}{\|\boldsymbol{X}\|_\infty} \leq \infty$.

4.(**Time-homogeneous**) $H = H_t : t \in \mathbb{R}$ is time-homogeneous (or shift-equivariant) if the input-output relationship commutes with time shift: let $[S_\tau(\boldsymbol{X})]_t = \boldsymbol{x}_{t-\tau}$ be a shift operator, then $H(S_\tau \boldsymbol{X}) = S_\tau H(\boldsymbol{X})$.

5.(**Causal**) $H_t$ is causal if it does not depend on future values of the input. That is, if $\boldsymbol{X}, \boldsymbol{X}'$ satisfy $\boldsymbol{x}_t = \boldsymbol{x}'_t$ for any $t \leq t_0$, then $H_t(\boldsymbol{X}) = H_t(\boldsymbol{X}')$ for any $t \leq t_0$.

6.(**Regular**) $H_t$ is regular if for any sequence $\boldsymbol{X}^{(n)} \in \mathcal{X} : n \in \mathbb{Z}$ such that $x_t^{(n)} \to \boldsymbol{0}$, then

$$\lim_{n \to \infty} H_t(\boldsymbol{X}^{(n)}) = \boldsymbol{0}.$$

According to the following lemma, the target we studied in Section 4 is a continuous, linear, regular, causal and time-homogeneous functional.

**Lemma E.1.** *Let* H *be a family of continuous, linear, regular, causal and time-homogeneous functionals on* $\mathcal{X}$. *Then, there exist a sequence* $\rho : \mathbb{N} \to \mathbb{R}^d$ *that is summable, i.e.*

$$\|\rho\|_{l^1} := \sum_{i=1}^{d} \sum_{j=0}^{\infty} |(\rho_i)_j| < \infty$$

*and*

$$H_t(\boldsymbol{X}) = \sum_{i=0}^{\infty} \rho_i \boldsymbol{x}_{t-i}, \ t \in \mathbb{Z}.$$

*In particular,* H *is uniformly bounded with* $\sup_t \|H_t\| = \|\rho\|_{\ell^1}$.

*Proof.* We prove the existence of a summable sequence $\rho : \mathbb{N} \to \mathbb{R}^d$ satisfying the stated properties. Let $\mathcal{X} = \ell^\infty(\mathbb{Z}; \mathbb{R}^d)$ with the sup-norm $\|\boldsymbol{X}\| = \sup_{k \in \mathbb{Z}} \max_{1 \leq i \leq d} |X_{k,i}|$.

**Step 1: Riesz representation and causality.**
Since each $H_t : \mathcal{X} \to \mathbb{R}$ is continuous, linear, and regular, by the Riesz representation theorem for $\ell^\infty$ spaces with regular functionals, there exists a unique *finitely additive* signed vector measure $\boldsymbol{\mu}_t$ on $\mathbb{Z}$ such that

$$H_t(\boldsymbol{X}) = \int_{\mathbb{Z}} \boldsymbol{X}^T d\boldsymbol{\mu}_t = \sum_{k \in \mathbb{Z}} \boldsymbol{X}_k^T \boldsymbol{\mu}_t(\{k\}),$$

where $\boldsymbol{\mu}_t(\{k\}) \in \mathbb{R}^d$ is the mass at integer $k$, and $\|H_t\| = \|\boldsymbol{\mu}_t\| = \sum_{k \in \mathbb{Z}} \sum_{i=1}^{d} |\mu_{t,i}(\{k\})|$.

Causality implies that $H_t(\boldsymbol{X})$ depends only on $\{\boldsymbol{X}_k : k \leq t\}$. Hence, $\boldsymbol{\mu}_t(\{k\}) = \boldsymbol{0}$ for all $k > t$, and the representation simplifies to:

$$H_t(\boldsymbol{X}) = \sum_{k=-\infty}^{t} \boldsymbol{X}_k^T \boldsymbol{\mu}_t(\{k\}). \tag{1}$$

**Step 2: Time homogeneity and shift invariance.**
For any $\tau \in \mathbb{Z}$, time homogeneity implies $H_t(\boldsymbol{X}) = H_{t+\tau}(\boldsymbol{X}^{(\tau)})$ where $\boldsymbol{X}_k^{(\tau)} = \boldsymbol{X}_{k-\tau}$. Applying (1) to the right-hand side:

$$H_{t+\tau}(\boldsymbol{X}^{(\tau)}) = \sum_{m=-\infty}^{t+\tau} \boldsymbol{X}_{m-\tau}^T \boldsymbol{\mu}_{t+\tau}(\{m\}).$$

Substituting $j = m - \tau$:

$$H_{t+\tau}(\boldsymbol{X}^{(\tau)}) = \sum_{j=-\infty}^{t} \boldsymbol{X}_j^T \boldsymbol{\mu}_{t+\tau}(\{j + \tau\}).$$

Equating with (1) and comparing coefficients:

$$\boldsymbol{\mu}_t(\{j\}) = \boldsymbol{\mu}_{t+\tau}(\{j + \tau\}) \quad \forall j \leq t. \tag{2}$$

**Step 3: Construction of $\rho$.**
Fix $t = 0$ in (2) and set $\tau = -t$:

$$\boldsymbol{\mu}_t(\{j\}) = \boldsymbol{\mu}_0(\{j + t\}) \quad \forall j \leq t.$$

Define $\boldsymbol{\mu}(\{k\}) = \boldsymbol{\mu}_0(\{k\})$. By causality, $\boldsymbol{\mu}(\{k\}) = \mathbf{0}$ for $k > 0$. Using time homogeneity:

$$\mathrm{H}_t(\boldsymbol{X}) = \mathrm{H}_0(\boldsymbol{X}^{(-t)}) = \sum_{k=-\infty}^{0} \boldsymbol{X}_{k+t}^T \boldsymbol{\mu}(\{k\}).$$

Substituting $i = -k$:

$$\mathrm{H}_t(\boldsymbol{X}) = \sum_{i=0}^{\infty} \boldsymbol{X}_{t-i}^T \boldsymbol{\mu}(\{-i\}). \tag{3}$$

Define $\rho_i = \boldsymbol{\mu}(\{-i\})$ for $i \geq 0$. Then (3) becomes:

$$\mathrm{H}_t(\boldsymbol{X}) = \sum_{i=0}^{\infty} \boldsymbol{x}_{t-i}^T \rho_i.$$

**Step 4: Summability and norm equality.**
From the Riesz representation and causality:

$$\|\rho\|_{\ell^1} = \sum_{i=0}^{\infty} \sum_{j=1}^{d} |(\rho_i)_j| = \sum_{k=-\infty}^{0} \sum_{j=1}^{d} |\mu_j(\{k\})| = \|\boldsymbol{\mu}\| = \|H_0\| < \infty.$$

For any $t \in \mathbb{Z}$, using (2) and shift invariance:

$$\|\mathrm{H}_t\| = \sum_{j=-\infty}^{t} \sum_{i=1}^{d} |\mu_{t,i}(\{j\})| = \sum_{j=-\infty}^{t} \sum_{i=1}^{d} |\mu_i(\{j + t\})| = \|\boldsymbol{\mu}\| = \|\rho\|_{\ell^1}.$$

Thus $\sup_{t \in \mathbb{Z}} \|\mathrm{H}_t\| = \|\rho\|_{\ell^1}$. $\qquad \square$

We now consider a continuous time-homogeneous system. According to Theorem C.6 in Wang & Li (2023) and de Figueiredo (1982), we obtain the following decomposition result for such systems:

**Lemma E.2** (Volterra Series Decomposition). H *is a continuous time-homogeneous system with input $\boldsymbol{X}$ and output $\boldsymbol{Y}$, then* H *can be expanded in the Volterra series as follows*

$$\boldsymbol{y}_t = \mathrm{H}_t(\boldsymbol{X}) = h_0 + \sum_{n=1}^{\infty} \sum_{\tau_1=0}^{\infty} \cdots \sum_{\tau_n=0}^{\infty} \mathrm{H}^{(n)}(\tau_1, \ldots, \tau_n)\big(\boldsymbol{x}_{t-\tau_1} \otimes \cdots \otimes \boldsymbol{x}_{t-\tau_n}\big), \tag{8}$$

*where $\otimes$ denotes the Kronecker product. In particular, we call the expansion order $n$ to be the series' order.*

We focus on the term $\mathrm{H}^{(n)}$ and derive its approximation error when using a single-layer Transformer to approximate it. We begin by stating the lemmas required for our analysis:

**Lemma E.3.** *Let $\{x_i\}_{i=1}^n$ be i.i.d. random variables with $\mathbb{E}[x_i] = 0$ and $\mathbb{E}[x_i^2] = 1$. Then $\bigotimes_{i=1}^n x_i$ is a random vector with mean zero and identity covariance matrix.*

*Proof.* It suffices to consider the scalar case $d = 1$. Define $y := \prod_{i=1}^n x_i$. Then

$$\mathbb{E}[y] = \mathbb{E}\left[\prod_{i=1}^n x_i\right] = \prod_{i=1}^n \mathbb{E}[x_i] = 0,$$

and

$$\mathbb{E}[y^2] = \mathbb{E}\left[\left(\prod_{i=1}^n x_i\right)^2\right] = \prod_{i=1}^n \mathbb{E}[x_i^2] = 1.$$

Thus $\mathrm{Var}(y) = 1$, which proves the claim. $\qquad \square$

**Lemma E.4.** $x_i$ *are random vectors with* $\mathbb{E}\|x_i\|_2 = B_i$, *and* $\|x_i - \bar{x}_i\|_2 \leq \varepsilon_i \leq \varepsilon$. *Then* $\mathbb{E}\|\bigotimes_{i=1}^n \tilde{x}_i - \bigotimes_{i=1}^n \hat{x}_i\|_2 \leq \prod_{i=1}^n (B_i + \varepsilon_i) - \prod_{i=1}^n B_i$

*Proof.* From the property of projection operation,

$$\|\tilde{x}_i - \hat{x}_i\|_2 \leq \|P\|_2 \|x_i - \hat{x}_i\|_2 \leq \varepsilon_i \leq \varepsilon.$$

By direct calculation,

$$\mathbb{E}\|\bigotimes_{i=1}^n \tilde{x}_i - \bigotimes_{i=1}^n \hat{x}_i\|_2 \leq \sum_{\phi \neq S \subset \{1,\ldots,n\}} \left(\prod_{i \in S} \varepsilon_i\right) \mathbb{E}\|\bigotimes_{j \notin S} \tilde{x}_j\|_2$$

$$= \sum_{\phi \neq S \subset \{1,\ldots,n\}} \left(\prod_{i \in S} \varepsilon_i\right) \prod_{j \notin S} B_j$$

$$= \prod_{i=1}^n (B_i + \varepsilon_i) - \prod_{i=1}^n B_i.$$

$\square$

We use the FFN to implement the Kronecker product in our proof; therefore, the universal approximation property of FFNs is also required.

**Definition E.2** (Barron space (E et al., 2019; 2021; Ma et al., 2020)). Consider functions $f : X \to \mathbb{R}$ that admit the following representation: $f(x) = \int_\Omega a\sigma(b^\top x + c)\rho(\mathrm{d}a, \mathrm{d}b, \mathrm{d}c)$, $x \in X$. For any $p \in [1, +\infty]$, we define the Barron norm as $\|f\|_{\mathcal{B}_p} := \inf_\rho \left(\mathbb{E}_\rho\left[|a|^p(\|b\|_1 + |c|)^p\right]\right)^{1/p}$. Then the Barron space are defined as: $\mathcal{B}_p := \{f \in \mathcal{C} : \|f\|_{\mathcal{B}_p} < +\infty\}$.

**Lemma E.5** (Ma et al. (2020)). *For any* $f \in \mathcal{B}$, *there exists a two-layer ReLU neural network* $\mathrm{FFN}(x) = \sum_{k=1}^K a_k \sigma(b_k^\top x + c_k)$ *with $M$ neurons such that*

$$\|f - \mathrm{FFN}\|_{L^\infty[-2,2]} \leq \mathcal{O}\left(\frac{\|f\|_{\mathcal{B}}\sqrt{\log K}}{\sqrt{K}}\right).$$

We now present the approximation and tradeoff result:

**Theorem E.1** (Tradeoff for Nonlinear Target). *To approximate the target* $\mathrm{H}_t(X) = \sum_{\tau_1=0}^\infty \cdots \sum_{\tau_n=0}^\infty \mathrm{H}^{(n)}(\tau_1, \ldots, \tau_n)(x_{t-\tau_1} \otimes \cdots \otimes x_{t-\tau_n})$, *we employ $M$ groups of heads, where group $m$ contains $H_m$ heads each of dimension $d_m \leq d$. Given the model dimension $D = \sum_{m=1}^M H_m \cdot d_m$ is fixed, with probability at least $1 - \delta$, we have*

$$\mathcal{E}_D(X) \leq \sum_{\tau_1=0}^M \cdots \sum_{\tau_n=0}^M \|\mathrm{H}^{(n)}(\tau_1, \ldots, \tau_n)\|_2 (1 + \varepsilon_\delta)^n \mathbb{I}_{\{\min_m d_m \leq d\}} \sqrt{1 - \left(\frac{\min_m d_m}{d}\right)^n}$$

$$+ \sum_{\tau_1=0}^M \cdots \sum_{\tau_n=0}^M \|\mathrm{H}^{(n)}(\tau_1, \ldots, \tau_n)\|_2 \left[(B + \varepsilon_{\mathrm{Attn}})^n - B^n\right]$$

$$+ \varepsilon_{\mathrm{H}} + \varepsilon_{\mathrm{FFN}}.$$

*where*

$$\varepsilon_\delta = \sqrt{\frac{2\log(2ML/\delta)}{\min_m d_m}},$$

$$\varepsilon_{\mathrm{Attn}} = \max_m \frac{1.3\, e^{0.02T}}{H_m},$$

$$\varepsilon_{\mathrm{H}} = \left\| \sum_{\tau_1=0}^\infty \cdots \sum_{\tau_n=0}^\infty \mathrm{H}^{(n)}(\tau_1, \ldots, \tau_n)(x_{t-\tau_1} \otimes \cdots \otimes x_{t-\tau_n}) \right.$$

$$\left. - \sum_{\tau_1=0}^M \cdots \sum_{\tau_n=0}^M \mathrm{H}^{(n)}(\tau_1, \ldots, \tau_n)(x_{t-\tau_1} \otimes \cdots \otimes x_{t-\tau_n}) \right\|_2.$$

*and $\varepsilon_{\mathrm{FFN}}$ is caused by using FFN to implement Kronecker product.*

**Corollary E.1** (Parameter allocation via trade-offs). *Under the same condition of Theorem E.1, the allocation of parameters can be achieved by solving the following optimization problem*

$$\min_{M,\,H_m,\,d_m} \sum_{\tau_1=0}^{M} \cdots \sum_{\tau_n=0}^{M} \left\| \mathrm{H}^{(n)}(\tau_1,\ldots,\tau_n) \right\|_2$$

$$\times \left[ \mathbb{I}_{\{\min_m d_m \leq d\}} \sqrt{1 - \left( \tfrac{\min_m d_m}{d} \right)^n} + (B + \varepsilon_{\mathrm{Attn}})^n - B^n \right] \tag{9}$$

$$s.t. \quad \sum_{m=1}^{M} H_m \cdot d_m = D.$$

*Proof.* In this proof, the construction of the embedding layer and the multi-head attention layer follows exactly the same procedure as in Theorem 4.1. After these two steps, for each token, we obtain:

$$\begin{pmatrix} \tilde{\boldsymbol{x}}_t \\ 0 \\ \vdots \\ 0 \end{pmatrix} \longrightarrow \begin{pmatrix} \tilde{\boldsymbol{x}}_t \\ \tilde{\boldsymbol{x}}_{t-1} \\ \vdots \\ \tilde{\boldsymbol{x}}_{t-M} \\ \boldsymbol{0}_{D-(M+1)\times d_H} \end{pmatrix} := \boldsymbol{x}_t^{(1/2)}.$$

By Lemma B.2, with probability at least $1 - M \max_m 2 \exp\left( -\frac{d_m \varepsilon^2}{2} \right)$, we have

$$\begin{aligned}
\mathcal{E}_D(\boldsymbol{X}) = & \Big\| \sum_{\tau_1=0}^{\infty} \cdots \sum_{\tau_n=0}^{\infty} \mathrm{H}^{(n)}(\tau_1,\ldots,\tau_n)\big( \boldsymbol{x}_{t-\tau_1} \otimes \cdots \otimes \boldsymbol{x}_{t-\tau_n} \big) \\
& - \sum_{\tau_1=0}^{M} \cdots \sum_{\tau_n=0}^{M} \tilde{\mathrm{H}}^{(n)}(\tau_1,\ldots,\tau_n)\big( \hat{\boldsymbol{x}}_{t-\tau_1} \otimes \cdots \otimes \hat{\boldsymbol{x}}_{t-\tau_n} \big) \Big\|_2 \\
\leq & \Big\| \sum_{\tau_1=0}^{\infty} \cdots \sum_{\tau_n=0}^{\infty} \mathrm{H}^{(n)}(\tau_1,\ldots,\tau_n)\big( \boldsymbol{x}_{t-\tau_1} \otimes \cdots \otimes \boldsymbol{x}_{t-\tau_n} \big) \\
& - \sum_{\tau_1=0}^{M} \cdots \sum_{\tau_n=0}^{M} \tilde{\mathrm{H}}^{(n)}(\tau_1,\ldots,\tau_n)\big( \boldsymbol{x}_{t-\tau_1} \otimes \cdots \otimes \boldsymbol{x}_{t-\tau_n} \big) \Big\|_2 \\
& + \Big\| \sum_{\tau_1=0}^{M} \cdots \sum_{\tau_n=0}^{M} \tilde{\mathrm{H}}^{(n)}(\tau_1,\ldots,\tau_n)\big( \tilde{\boldsymbol{x}}_{t-\tau_1} \otimes \cdots \otimes \tilde{\boldsymbol{x}}_{t-\tau_n} \big) \\
& - \sum_{\tau_1=0}^{M} \cdots \sum_{\tau_n=0}^{M} \tilde{\mathrm{H}}^{(n)}(\tau_1,\ldots,\tau_n)\big( \tilde{\boldsymbol{x}}_{t-\tau_1} \otimes \cdots \otimes \tilde{\boldsymbol{x}}_{t-\tau_n} \big) \Big\|_2 \\
& + \Big\| \sum_{\tau_1=0}^{M} \cdots \sum_{\tau_n=0}^{M} \tilde{\mathrm{H}}^{(n)}(\tau_1,\ldots,\tau_n)\big( \tilde{\boldsymbol{x}}_{t-\tau_1} \otimes \cdots \otimes \tilde{\boldsymbol{x}}_{t-\tau_n} \big) \\
& - \sum_{\tau_1=0}^{M} \cdots \sum_{\tau_n=0}^{M} \tilde{\mathrm{H}}^{(n)}(\tau_1,\ldots,\tau_n)\big( \hat{\boldsymbol{x}}_{t-\tau_1} \otimes \cdots \otimes \hat{\boldsymbol{x}}_{t-\tau_n} \big) \Big\|_2 \\
\leq & \sum_{\tau_1=0}^{M} \cdots \sum_{\tau_n=0}^{M} \left\| \mathrm{H}^{(n)}(\tau_1,\ldots,\tau_n) \right\|_2 (1+\varepsilon)^n \mathbb{I}_{\{\min_m d_m \leq d\}} \sqrt{1 - \left( \frac{\min_m d_m}{d} \right)^n} \\
& + \sum_{\tau_1=0}^{M} \cdots \sum_{\tau_n=0}^{M} \left\| \mathrm{H}^{(n)}(\tau_1,\ldots,\tau_n) \right\|_2 \left[ (B + \varepsilon_{\mathrm{Attn}})^n - B^n \right] + \varepsilon_{\mathrm{H}} + \varepsilon_{\mathrm{FFN}}
\end{aligned}$$

where $\varepsilon_{\text{Attn}} = \max_m \frac{1.3\, e^{0.02T}}{\text{H}_m}$, $\varepsilon_H = \|\sum_{\tau_1=0}^{\infty} \cdots \sum_{\tau_n=0}^{\infty} \text{H}^{(n)}(\tau_1,\ldots,\tau_n)(\boldsymbol{x}_{t-\tau_1} \otimes \cdots \otimes \boldsymbol{x}_{t-\tau_n}) - \sum_{\tau_1=0}^{M} \cdots \sum_{\tau_n=0}^{M} \tilde{\text{H}}^{(n)}(\tau_1,\ldots,\tau_n)(\hat{\boldsymbol{x}}_{t-\tau_1} \otimes \cdots \otimes \hat{\boldsymbol{x}}_{t-\tau_n})\|_2$, and $\varepsilon_{\text{FFN}}$ is caused by using FFN to implement Kronecker product.

Take uniform bound over $L$ tokens, with probability at least $1 - \delta$, we have

$$
\mathcal{E}_D(\boldsymbol{X}) \leq \sum_{\tau_1=0}^{M} \cdots \sum_{\tau_n=0}^{M} \left\|\text{H}^{(n)}(\tau_1,\ldots,\tau_n)\right\|_2 (1 + \varepsilon_\delta)^n \mathbb{I}_{\{\min_m d_m \leq d\}} \sqrt{1 - \left(\frac{\min_m d_m}{d}\right)^n}
$$
$$
+ \sum_{\tau_1=0}^{M} \cdots \sum_{\tau_n=0}^{M} \left\|\text{H}^{(n)}(\tau_1,\ldots,\tau_n)\right\|_2 \left[(B + \varepsilon_{\text{Attn}})^n - B^n\right]
$$
$$
+ \varepsilon_H + \varepsilon_{\text{FFN}}.
$$

where $\varepsilon_\delta = \sqrt{\frac{2 \log (2ML/\delta)}{\min_m d_m}}$. $\qquad\square$

# F  ADDITIONAL EXPERIMENTS

**Inspiration of parameter allocation.**   Jiang et al. (2025) empirically investigated how the head trade-off is influenced by the memory structure and the difficulty of the target relationship in Figure 8. As illustrated in Figure 4 of their work, the trade-off depends on both the underlying memory structure and the level of task difficulty. Each subfigure corresponds to a specific memory structure, with the plots showing loss as a function of the number of heads. The color of the curves encodes the parameter $\alpha$, which reflects task difficulty. The results indicate that, for certain tasks, the trade-off remains consistent across different levels of difficulty (Figure (c)). In tasks (a) and (b), when the task is relatively easy, no clear trade-off is observed. However, in task (c), a trade-off appears consistently, regardless of task difficulty.

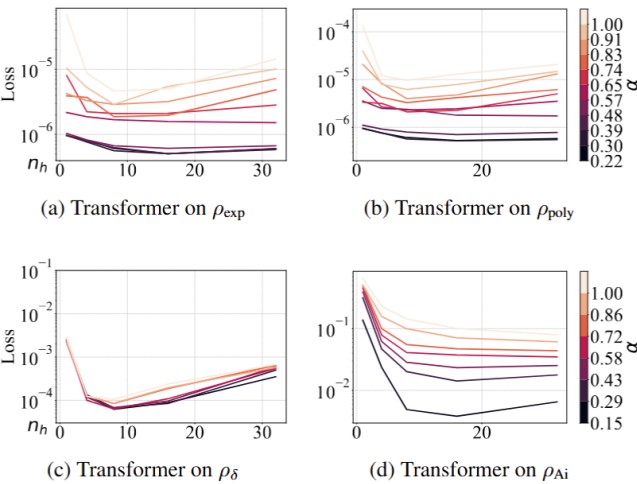

(a) Transformer on $\rho_{\text{exp}}$          (b) Transformer on $\rho_{\text{poly}}$

(c) Transformer on $\rho_\delta$          (d) Transformer on $\rho_{\text{Ai}}$

Figure 4: Loss versus $n_h$ for Transformers with fixed $m = 128$ and varying $\alpha$.

Figure 8: Linear Convolution Tradeoff

**Reducing budget on number of heads.**   We conducted a systematic investigation of Transformer architectural hyperparameters on the WikiText-103 language modeling task. The goal was to understand how attention head dimension and number of heads affect model performance in next-token prediction.

The models were 6-layer GPT-style Transformer decoders with learned token and positional embeddings (sequence length 256), RMSNorm, SwiGLU feed-forward networks, residual connections, and weight-tied output projections. Two hyperparameter sweeps were performed: (1) varying head dimension with 8 attention heads fixed, and (2) varying the number of heads (1–32) with head dimension fixed at 64. The model dimension $d_{\text{model}}$ was set as the product of number of heads and head dimension.

Training was performed on $1\%$ of WikiText-103 ( 1M tokens) for 2 epochs using AdamW (learning rate $3 \times 10^{-4}$, weight decay 0.1) with cosine scheduling, 200-step warmup, gradient clipping, dropout 0.1, and mixed precision. Models were evaluated on validation and test sets using cross-entropy loss and perplexity. Each configuration recorded training curves, validation losses, and test performance for analysis.

Comparative analysis reveals that cutting head dimension is a more effective way to reduce model parameters than reducing the number of heads, which incurs higher loss. The study highlights how architectural choices influence performance and parameter efficiency in language modeling, offering insights for balancing model capacity and computational budget.

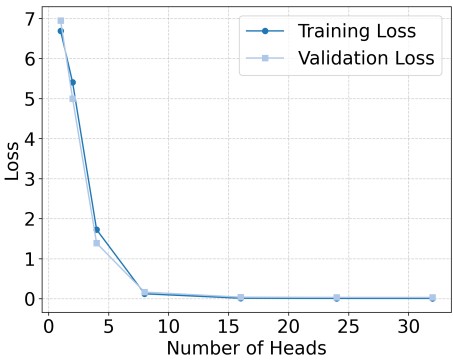

Figure 9: Loss vs Number of Heads

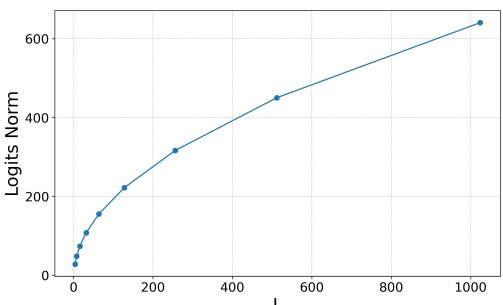

Figure 10: Logits Norm vs L

**Analysis of attention Logits norms.** In Figure 10, we investigate how the $\ell_2$ norm of attention logits (pre-softmax) varies with sequence length. For each sequence length $L$, we randomly sampled 10 text sequences from the Wikitext-103 dataset and performed forward passes using the `TinyLlama-1.1B model`. We focused on the logits of the first attention head in the first layer (layer 0, head 0) and computed the $\ell_2$ norm at the middle token of each sequence (index $= L/2$).

This analysis has several theoretical motivations:

- **Gradient stability:** The magnitude of logits directly affects the softmax gradients. Excessively large norms may cause gradient explosion, while very small norms lead to overly uniform attention distributions.

- **Numerical stability:** Extreme logits values can result in numerical overflow, impacting training stability.

- **Attention concentration:** Differences in logits magnitudes indicate the degree of attention focus, with larger variations corresponding to more concentrated attention.

By analyzing the logits norm at the middle token, we avoid boundary effects and ensure comparability across sequences of different lengths. We hypothesize that the logits norm exhibits a specific scaling behavior with sequence length and should remain relatively stable to support effective gradient propagation. These observations provide insights into the internal mechanisms of Transformers when handling long sequences and can guide design choices for long-sequence modeling.

