# OpenReview forum: "Allocation of Parameters in Transformers"
_ICLR.cc/2026/Conference — ICLR 2026 Conference Withdrawn Submission_

### Official Review · Reviewer_EnX5 · 2025-10-30

**Soundness:** 2
**Presentation:** 4
**Contribution:** 3
**Rating:** 4
**Confidence:** 4

**Summary:**

This paper establishes a theoretical foundation for Transformer model efficiency in two directions. First, it provides mathematical analysis of how the number of heads and head dimensions affect approximation error in early layers, characterizing the trade-off under a fixed parameter budget. Second, it analyzes softmax saturation with respect to sequence length, demonstrating diminishing returns from increasing head dimensions and motivating parameter reduction in later layers. Experimental results with several Transformer variants validate these theoretical findings.

**Strengths:**

- The theoretical framework connecting approximation analysis and softmax saturation to compression strategies is well-motivated and provides principled insights for parameter-efficient Transformer design.
- The paper is well written and easy to follow
- The paper validates theoretical findings across multiple experimental settings, including synthetic tasks, n-gram prediction, and pre-trained model compression, demonstrating the generality of the observed phenomena.

**Weaknesses:**

- Lack of a clear connection between empirical analysis by Chen et al., (2024) and the proposed dimension compression derived from saturation pattern in softmax activations. The paper cites Chen et al., (2024) to argue that later layers empirically activate only few heads, suggesting that softmax saturation particularly enables efficient parameter reduction in those layers. However, the connection between these two observations remains unclear. In addition, the paper does not provide a clear theoretical or empirical link explaining why dimension reduction in later layers is more effective than in early layers. Specifically, Corollary 5.1 does not include layer dependencies, nor do the experiments in Section 6.2 provide sufficient evidence to support the claim of superior compression efficiency in later layers.
- Insufficient evaluation of whole-model effects. The experiments in Figures 4 and 6 only replace a single attention head with a reduced-dimension variant, without evaluating the impact when such compression is applied across all heads and layers simultaneously. This remains unclear how the compression effects on the model performance. Moreover, the paper lacks both experimental and theoretical guidance on how to apply the proposed strategies in practice.
- Lack of clear definitions for "early" and "later" layers. The paper proposes different strategies for early versus later layers but does not specify how to identify this boundary (e.g., at which layer index in a 6-layer, 12-layer, or 24-layer model). This ambiguity makes it difficult to apply the proposed strategies in practice.

**Questions:**

- Does the softmax saturation pattern differ across layers? Is saturation
more severe in later layers?
- Can the proposed compression be applied to entire models while maintaining
performance?
- Is the proposed approach compatible with existing efficiency techniques
like Grouped Query Attention (GQA)? Can these methods be combined?

---

### Official Review · Reviewer_Hf2G · 2025-10-30

**Soundness:** 2
**Presentation:** 2
**Contribution:** 1
**Rating:** 2
**Confidence:** 4

**Summary:**

This paper investigates the parameter efficiency of transformer models. The author analyzes how parameter allocation, particularly in terms of head dimension and hidden dimension, influences different layers within the model.

**Strengths:**

1. This paper provides a quantitative analysis of the parameter allocation efficiency.

2. The paper is well-organized.

**Weaknesses:**

There are several concerns about this paper.
1. The first is that the contribution appears quite incremental. The author examines the parameter efficiency of transformer models, specifically by fixing the attention parameters (i.e., keeping the product of the number of heads and head dimension constant). However, the primary unresolved bottleneck of transformers lies in the input length $L$,  rather than in the hidden dimension or number of heads alone. Moreover, many prior works have already explored the trade-off between head count and head dimension, dating back to as early as 2019 [1]. As a result, it is unclear what new insights this paper contributes beyond existing research.

2. Second, the finding that there exists a trade-off between these parameters is not novel. Previous studies have already demonstrated the low-rank nature of attention mechanisms[2], parameter efficiency[3][4] . While this paper provides theoretical support for parameter allocation in early layers from an approximation theory perspective, similar empirical and theoretical evidence already exists in the literature . Although this approximation-theoretic view offers an alternative angle, it does not appear to introduce substantially new or inspiring insights.

3.Third, the experimental section is quite limited. When exploring layer-related properties, validating results only on small transformer architectures is not sufficiently convincing, as the findings may not hold when the model scale increases significantly. Also, why the theories demonstrated in the paper are useful are not shown in the paper.


Reference:
[1]Michel, Paul, Omer Levy, and Graham Neubig. "Are sixteen heads really better than one?." Advances in neural information processing systems 32 (2019).

[2]Bhojanapalli, Srinadh, et al. "Low-rank bottleneck in multi-head attention models." International conference on machine learning. PMLR, 2020.

[3]Liu, Xiaoyu, Jiahao Su, and Furong Huang. "Tuformer: Data-driven design of transformers for improved generalization or efficiency." The Tenth International Conference on Learning Representations (ICLR 2022). 2022.

[4]Panahi, Aliakbar, Seyran Saeedi, and Tom Arodz. "Shapeshifter: a parameter-efficient transformer using factorized reshaped matrices." Advances in Neural Information Processing Systems 34 (2021): 1337-1350.

**Questions:**

1. When the model scale increases, does the proposed theory still hold?

2. In the sequence modeling task, there is an implicit assumption that information can be extracted from M tokens. Why, then, should each group be dedicated to a single token?

3. Corollary 5.1 primarily states that there exists a low-rank approximation of the original construction. Is there any experimental evidence showing that, with your truncated SVD approximation, the downstream task performance remains at a comparable level?

---

### Official Review · Reviewer_9NtQ · 2025-11-01

**Soundness:** 2
**Presentation:** 3
**Contribution:** 2
**Rating:** 4
**Confidence:** 3

**Summary:**

This paper talks about the efficient way to allocate attention heads and dimensions across transformer layers in both theoretical and practical ways. There are two main theoretical contributions: 1) an approximation error estimate on information extraction in early transformer layers implies a trade-off between number of heads and head dimensions under fixed parameter budgets; 2) a softmax saturation theorem shows that continuously increasing head dimensions leads to diminishing returns, especially for long sequences. They provide analysis of pretrained models to verify the theoretical findings, including synthetic tasks and single head compression experiments on pre-trained models.

**Strengths:**

The approximation theory analysis of parameter allocation trade-offs in transformers is well motivated and quite novel. Especially Section 5 provides a new standpoint for softmax attention about the saturation. The overall theoretical framework is mathematically rigorous, and are mostly supported by the experiments.

**Weaknesses:**

1. Section 4 proves the trade-offs in information extraction in transformer layers, however, the linear sequence modeling framework is overly restrictive and the nonlinear sequence modeling also has assumptions that are hard to achieve. The authors don’t have an explanation on why single layer analysis can be representative for real language modeling tasks which is highly nonlinear, or why these theorems are also applicable to multi-layer models. While the theory on softmax saturation in Section 5 is also on single-layer transformer, the experiments are done in pretrained multi-layer models, also without proper connections.
2. The experimental validation is overall very simple. There is no comparisons with other standard architectural choices, e.g., uniform head allocation. And no other metrics are analyzed besides training and validation loss.

**Questions:**

1. Theorem 5.1 assumes bounded logits with O(1/L) but Figure 10 shows logits norms grow with sequence length L, does this invalidate the previous theory?’
2. How will the trade-off changes across different layers in a transformer?

---

### Official Review · Reviewer_KCZo · 2025-11-04

**Soundness:** 3
**Presentation:** 2
**Contribution:** 3
**Rating:** 4
**Confidence:** 4

**Summary:**

The paper studies how to allocate attention heads and head dimensions across Transformer layers under a fixed parameter budget. The authors argue that early layers require more parameters for information extraction, leading to a head-vs-dimension trade-off. In contrast, later layers can afford parameter reduction due to "softmax saturation," where increasing d_h yields diminishing returns, especially for long sequences. Empirical validations utilise synthetic tasks and small pre-trained models on datasets such as WikiText-103, demonstrating trade-offs and feasibility of compression.

**Strengths:**

- The paper tackles an under-explored area of parameter allocation (not just pruning).

- The paper offers a clean mathematical framing of parameter trade-offs in early layers via approximation theory.

- These theoretical claims are well-supported by the experiments in Section 6.2 and Figures 3-6.

**Weaknesses:**

- Some of the ideas mentioned in this paper as novel are not a discovery by this paper. For example the the softmax saturation behaviour is mentioned in the related work which cites Mudarisov et al. (2025), who show that "as sequence length grows, attention weights collapse toward uniformity" and "token separability saturates". Besides,


- Experiments are toy-scale: Synthetic 4-gram on white noise, single-head compression in models, and 6-layer decoders on 1% WikiText-103 (~1M tokens, Fig. 5). No large-scale results (e.g., on LLaMA-7B+), no downstream tasks (e.g., GLUE, long-context benchmarks like RULER), and no latency/FLOPs measurements despite efficiency claims.

- The theoretical model of early-layer function is too simple, and its central experimental validation (Fig 2) directly contradicts its theoretical prediction (Eq. 3). Can the authors clarify on this?

**Questions:**

- For a simple 4-gram task, the authors' math (Eq. 3) predicts that the ideal setup is 32 attention heads. In the Experiment, when they actually run the experiment (Figure 2), the best performance clearly comes from using only 8 heads. How is this contradiction happening?

**Details Of Ethics Concerns:**

N/A.

---

### Note · Authors · 2025-11-17

**Comment:**

We would like to withdraw our submission due to issues identified in the current version. We appreciate the reviewers’ time and consideration.

**Withdrawal Confirmation:**

I have read and agree with the venue's withdrawal policy on behalf of myself and my co-authors.